

# Adaptive Baseline Finder, a statistical data selection strategy to identify atmospheric $CO_2$ baseline levels and its application to European elevated mountain stations

Ye Yuan[1], Ludwig Ries[2], Hannes Petermeier[3], Martin Steinbacher[4], Angel J. Gómez-Peláez[5], Markus C. Leuenberger[6], Marcus Schumacher[7], Thomas Trickl[8], Cedric Couret[2], Frank Meinhardt[9], Annette Menzel[1,10]

[1]Department of Ecology and Ecosystem Management, Technische Universität München, Freising, Germany
[2]German Environment Agency (UBA), Zugspitze, Germany
[3]Department of Mathematics, Technische Universität München, Freising, Germany
[4]Empa, Laboratory for Air Pollution/Environmental Technology, Dübendorf, Switzerland
[5]Izaña Atmospheric Research Center, Meteorological State Agency of Spain (AEMET), Santa Cruz de Tenerife, Spain
[6]Climate and Environmental Physics Division, Physics Institute and Oeschger Centre for Climate Change Research, University of Bern, Bern, Switzerland
[7]Meteorological Observatory Hohenpeissenberg, Deutscher Wetterdienst (DWD), Hohenpeissenberg, Germany
[8]Institute of Meteorology and Climate Research, Atmospheric Environmental Research (IMK-IFU), Karlsruhe Institute of Technology (KIT), Garmisch-Partenkirchen, Germany
[9]German Environment Agency (UBA), Schauinsland, Germany
[10]Institute for Advanced Study, Technische Universität München, Garching, Germany

*Correspondence to*: Ye Yuan (yuan@wzw.tum.de)

**Abstract.** Critical data selection is essential for determining representative baseline levels of atmospheric trace gas measurements even at remote measuring sites. Different data selection techniques have been used around the world which could potentially lead to bias when comparing data from different stations. This paper presents a novel statistical data selection method based on $CO_2$ diurnal pattern occurring typically at high elevated mountain stations. Its capability and applicability was studied for atmospheric measuring records of $CO_2$ from 2010 to 2016 at six Global Atmosphere Watch (GAW) stations in Europe, namely Zugspitze-Schneefernerhaus (Germany), Sonnblick (Austria), Jungfraujoch (Switzerland), Izaña (Spain), Schauinsland (Germany) and Hohenpeissenberg (Germany). Three other frequently applied statistical data selection methods were implemented for comparison. Among all selection routines, the new method named Adaptive Baseline Finder (ABF) resulted in lower selection percentages with lower maxima during winter and higher minima during summer in the selected data. To investigate long-term trend and seasonality, seasonal decomposition technique STL was applied. Compared with the unselected data, mean annual growth rates of all selected data sets were not significantly different except for Schauinsland. However, clear differences were found in the annual amplitudes as well as for the seasonal time structure. Based on correlation analysis, results by ABF selection showed a better representation of the lower free tropospheric conditions.



# 1 Introduction

Continuous in situ measurements of greenhouse gases (GHG) at remote locations have been established since 1958 (Keeling, 1960). Knowledge of background atmospheric GHG concentrations is a key to the understanding of the global carbon cycle and its effect on climate, as well as the GHG responses to a changing climate. A crucial issue when using data from remote stations remains the identification of time periods that are representative for larger spatial areas and their separation from periods influenced by local and regional pollution. If these two regimes are well disaggregated, the available data sets can represent more reliable information about long-term changes of undisturbed atmospheric GHG levels, or investigate local and regional GHG sources and sinks when specifically analyzing the deviations from the baseline conditions. In this study, the baseline conditions refer to a selected subset of data from the validated data set, representing well-mixed air masses with minimized short-term external influences (Elliott, 1989; Calvert, 1990; Balzani Lööv et al., 2008; Chambers et al., 2016).

Measurement results depend on sampling methods, analytical instrumentation and data processing. Validated data (labelled as VAL in this study to differentiate from the selected data) are usually obtained after signal correction, for example due to interferences from other GHG, calibration accounting for sensitivity changes of the analyzer, and validation based on plausibility checks. Data selection starts with validated data and identifies in subsequent steps a final subset of the validated data set based on pre-defined criteria for a specific quality such as representativeness. With a particular focus on $CO_2$ in this study, it is commonly accepted that data selection methods can be categorized into meteorological, tracer and statistical selection methods (Ruckstuhl et al., 2012; Fang et al., 2015).

Meteorological data selection makes use of the meteorological information at the measuring site, which provides valuable information about the surrounding environment as well as air mass transport (Carnuth and Trickl, 2000; Carnuth et al., 2002). Forrer et al. (2000), Zellweger et al. (2003), and Kaiser et al. (2007) studied intensively the relationship between measured trace gases (such as $O_3$, CO and $NO_x$) and meteorological processes at Zugspitze, Jungfraujoch, Sonnblick and Hohenpeissenberg. For $CO_2$, the most common parameters applied in the literature are wind speed and wind direction. They can provide information on critical variations at stations with sources and sinks in their vicinity while these parameters are less suited at stations in largely pristine environments. For example, Lowe et al. (1979) performed a pre-selection on the $CO_2$ record at Baring Head (New Zealand) during the southerly wind period only (clean marine air). Massen and Beck (2011) found that the $CO_2$ versus wind speed plot can be valuable for baseline $CO_2$ estimation without local influence of continental measurements. Besides, fixed time window selection has been widely used, by selecting data in a certain time interval of the day based on local and mesoscale mechanisms of air mass transportation. For selecting well-mixed air at elevated mountain sites, night time is usually chosen with a special focus on the exclusion of afternoon periods (Bacastow et al., 1985). Brooks et al. (2012) also limited their mountaintop $CO_2$ results in the Rocky Mountains (USA) by "time-of-day" from 0 a.m. till 4 a.m. local time (LT) for a more likely free tropospheric environment sampled at the station. Apart from this, modeling techniques such as backward trajectories are very helpful for analyzing in detail the origins and transport processes of air masses arriving at the station (Cui et al., 2011; Uglietti et al., 2011). Using tracers, data selection can be performed by



investigating the correlations between the air components of interest. Many tracers have been applied and compared with $CO_2$. Threshold limits of 300 ppb for CO and 2000 ppb for $CH_4$ were defined to filter the analyzed $CO_2$ data at Lutjewad (the Netherlands) and Mace Head (Ireland) by Sirignano et al. (2010). Similar approaches with black carbon and $CH_4$ were performed by Fang et al. (2015) at Lin'an (China). Chambers et al. (2016) developed and applied a data selection technique to identify baseline air masses using atmospheric radon measurements at the stations Cape Grim (Australia), Mauna Loa (USA) and Jungfraujoch.

Unlike most of the methods mentioned above, which require additional data or advanced transport modelling, statistical data selection only relies on the time series of interest and typically investigates the variability of signal. It is usually assumed that the most representative $CO_2$ data are found during well-mixed conditions revealing small variations in time (Peterson et al., 1982) and in space (Sepúlveda et al., 2014). For continuous measurements, it is possible to investigate within-hour and hour-to-hour variability in the data sets. The within-hour variability is often expressed as the standard deviation of the measured data within one hour. The hour-to-hour variability compares the differences between hourly averaged concentrations either during a certain time period, or from one hour to the next. Pales and Keeling (1965) marked ambient data as "variable" when the within-hour variability for the air sample is significantly larger than the within-hour variability for the reference gas. Consequently they only selected $CO_2$ data in "steady" conditions for 6 hours or more. Besides, Peterson et al. (1982) also rejected sampled $CO_2$ data values for adjacent hours when the hour-to-hour variability exceeded 0.25 ppm. Thoning et al. (1989) combined these two strategies using an iterative approach by selecting data according to deviations of daily averages from a spline curve fit. Ruckstuhl et al. (2012) chose a different statistical method based on robust local regression (REBS) to estimate the baseline curves generalized for atmospheric compounds, which is available in the R package IDPmisc (Locher and Ruckstuhl, 2012).

The present study focuses on the comparison of results from statistical data selection methods and the Adaptive Baseline Finder (ABF). The ABF is seen as a possible alternative to already known data selection methods as discussed above. By applying ABF to the atmospheric $CO_2$ records from six European elevated mountain stations, the selection results are compared with those derived from three other statistical data selection methods. To investigate the potential influences of the data selection method on trend and seasonality, further analyses focus on the decomposition of validated and selected data sets in trend and seasonal components. Finally, differences between ABF and other data selection methods were assessed by correlation analysis.

## 2 Methods

### 2.1 $CO_2$ measurements at elevated European sites

$CO_2$ measurements from six selected European mountain stations (see Fig. 1) within the Global Atmosphere Watch (GAW) network were used to test the data selection algorithms. Three high alpine measuring sites were included: Zugspitze-Schneefernerhaus (*ZSF*, DE, 47°25' N, 10°59' E, 2670 m a.s.l.), Jungfraujoch (*JFJ*, CH, 46°33' N, 7°59' E, 3580 m a.s.l.)



and Sonnblick (*SNB*, AT, 47°03' N, 12°57' E, 3106 m a.s.l.). They are often above the planetary boundary layer (PBL), and thus exposed to free and assumed clean lower tropospheric air masses, but periodically influenced by regional emissions from lower altitudes. Additionally, to test data selection for a less remote environment, $CO_2$ measurements from Schauinsland (*SSL*, DE, 47°55' N, 7°55' E, 1205 m a.s.l.) at a clearly lower elevation in the mid-range Black Forest were

investigated. Data selection was also applied to three recently started $CO_2$ time series from different sampling heights above ground at a tall tower at the Hohenpeissenberg observatory (*HPB*, DE, 47°63' N, 11°01' E, 934 m a.s.l.), located in the Northern foothills of the Alps. Based on the station categorization by Henne et al. (2010), *JFJ* and *SNB* are classified as "mostly remote", while *ZSF* is considered as "weakly influenced, constant deposition", and *SSL* and *HPB* are considered as "rural". At last, Izaña station on Tenerife Island in the North Atlantic (*IZO*, ES, 28°19' N, 16°30' W, 2373 m a.s.l.) was

chosen as a reference for comparison due to its location above the subtropical temperature inversion layer which makes the station to be rarely affected by any local or regional $CO_2$ sources and sinks (Gomez-Pelaez et al., 2013).

The validated $CO_2$ hourly averages from all stations are available at the World Data Centre for Greenhouse Gases (WDCGG; http://ds.data.jma.go.jp/gmd/wdcgg/). Data with higher time resolution required for this study were provided directly by the station investigators. Descriptions of the sampling elevation and time period of available data are given in Table 1. Further

information on each station can be found in Schmidt et al. (2003), Gilge et al. (2010), Gomez-Pelaez et al. (2010), Risius et al. (2015) and Schibig et al. (2015). Practical data selections and analyses in this paper have been performed in the R Statistical Environment (R Core Team, 2017).

**2.2 Adaptive Baseline Finder (ABF)**

ABF is motivated by the investigation of $CO_2$ diurnal cycles at elevated mountain stations that are in the free troposphere or

representing a regional background during certain periods. Even though such measuring sites are remotely located, the $CO_2$ levels are still influenced by local sources and sinks. For instance, both traffic activities starting in the morning hours, and vegetation activities in the afternoon hours contribute to the measured $CO_2$ signal at *ZSF* on a diurnal basis. This points out the importance of finding a certain diurnal time window representing the most stable and representative $CO_2$ level which in turn is an effective tool for data selection. However, the duration of this time window during the day varies with seasons and

from day to day (e.g., due to changing degree of entrainment of PBL air). In summer, larger variabilities in the $CO_2$ signal are observed due to more prevalent convective boundary layer air mass injections influencing the diurnal pattern resulting in shorter periods of stable condition. In winter, much longer stable periods can be found. No upwind air masses with depleted $CO_2$ levels by photosynthesis of vegetation like in summer are recorded. To capture as many representative data as possible, it is desirable to select the time window dynamically. ABF is constructed to fulfil these requirements for an automated data

selection method that selects a subset of the validated data being best representative for baseline conditions at the measuring station with an adaptive selected time window specific for every day.

The algorithm is based on two basic assumptions. Firstly, air masses measured at altitude stations contain well-mixed air, closest to baseline levels, within a certain time window of several hours during the day. For the elevated mountain stations



discussed in this paper, this time interval is around midnight. Different diurnal patterns are apparent at each station so that the selected time window should be adjusted accordingly. Secondly, it is assumed that the real baseline conditions are not subject to local influences and thus represent air masses originated only from the uninfluenced lower free troposphere. It indicates that the variability of the measured $CO_2$ signal should be minimal within this selected time window. The

methodological steps of ABF are introduced in detail below in the two sections *starting selection* and *adaptive selection*.

**2.2.1 *Starting selection***

For a given validated data set, ABF starts data selection by finding a *start time window* for all days. The standardized selection procedure for the *start time window* results from site-specific parameters. This time interval is set as the most stable period from the diurnal variation. The step is referred to as *starting selection*. It begins by analyzing the mean diurnal cycle

of the data input.

**Step 1**: Detrending is done by subtracting a 3-day average for each day, including the neighboring two days. It is the shortest possible time window to remove sudden changes in the time series related to the previous and posterior days, but preserves of the diurnal pattern.

**Step 2**: The overall mean diurnal variation, $\bar{d}_i$ ($i = 0 \ to \ 23 \ h$), are calculated from the complete set of detrended data.

**Step 3**: The standard deviations $s_{\Delta_j}$ from the overall mean diurnal variation $\bar{d}_i$ are calculated on a moving window $\Delta_j$ ($j = 6 \ h$). To be able to place a full set of 24 moving time windows over the overall mean diurnal variation, time windows across midnight (e.g. 6 hours from 11 p.m. to 4 a.m. LT) are also included, i.e. its first $j$ hours are appended to the end of the 24 hours in the overall mean diurnal variation. The time window with the smallest standard deviation is selected as the *start time window*.

**Result**: The *start time window* $[i_{start}, ..., i_{end}]$.

With the focus on elevated mountain stations, *starting selection* is purposely designed with the moving window $\Delta_j$ of 6 hours, and the starting hour $i_{start}$ to be between 6 p.m. and 5 a.m. LT for this study. For other stations with possibly different diurnal patterns, *starting selection* can be adjusted accordingly. For instance, at urban stations or stations completely within the continental PBL, the *start time window* can be chosen based on the best mixing conditions that often occur in the

afternoon with a shorter moving window, when the PBL reaches its maximum depth after "ingesting" free-tropospheric air during its growth. Being aware that calculating the *start time window* from overall data could differ from the *start time windows* calculated by seasons, the overall generated *start time windows* have been compared with seasonal generated *start time windows* for high elevated mountain stations (see Supplement S1.1). Because these differences are mostly minimal to moderate and this work aims to a methodical comparison under identical conditions, a constant generation of *start time*

*window* from overall data has been chosen.



### 2.2.2 *Adaptive selection*

The second part *adaptive selection* is designed to determine the most suitable time window for each day, based on the data variability. The length of the *start time window* is adapted (expanding only) in both directions in time. *Adaptive selection* is done on a daily basis, starting with the first day of the given data set. The following steps only describe the *forward adaptive selection*. ABF runs the *backward adaptive selection* in an analogous manner reverse in time.

**Step 1**: The mean mole fraction $\bar{x}_i$, standard deviation $s_i$ and the proportion of missing values $\pi_{missing}$ are calculated from data in the *start time window* $[i_{start}, \ldots, i_{end}]$.

**Step 2**: If $s_i \leq 0.3$ ppm ($CO_2$) and $\pi_{missing} \leq 0.5$, ABF continues. Otherwise, it is considered that the *start time window* does not fulfill the assumptions. No data is selected for this day. Go to **Next Day**.

**Step 3**: ABF advances in time to examine whether the next data point $x_f$ can be included in the selected time window $W$ with $f = i_{end} + 1$.

**Step 4**: The absolute difference between $x_f$ and $\bar{x}_i$ is calculated, and the following threshold criterion is applied: $|x_f - \bar{x}_i| \leq \kappa \cdot s_i$, where $\kappa$ is the threshold parameter. If this criterion holds, $x_f$ is included in $W$ and ABF continues. Otherwise, ABF is finished only with the *start time window* for this day and go to **Next Day**.

**Step 5**: Mean $\bar{x}_W$ and standard deviation $s_W$ for the new selected time window $W$ are calculated. If $s_W \leq 0.3$ ppm ($CO_2$), ABF continues with next data point $x_f$ while $f = f + 1$. Otherwise, ABF is finished with the previous selected time window and go to **Next Day**.

**Step 6**: The new absolute difference between $x_f$ and $\bar{x}_W$ is calculated, as well as the new threshold criteria. If condition $|x_f - \bar{x}_W| \leq \kappa \cdot s_W$ holds, $x_f$ is included in $W$ and ABF goes back to **Step 5**. Otherwise, ABF is finished for this day and go to **Next Day**.

When selection for all days is finished, ABF continues with *backward adaptive selection*. Afterwards, go to **Result**.

**Result**: The final selected time window, which is a combination of $W_{forward}$ and $W_{backward}$ for the referring day.

The following limitations of the forward and backward expansions of the time window should be considered. ABF always runs no longer than 24 hours including the *start time window*, i.e. $f \leq 24 \cdot tr$, where $tr$ is the time resolution in data points per hour of the input data. Sometimes this results in an overlap of "selected" and "unselected" data for two consecutive days. We always consider the data as "selected" once it has been selected by ABF in any day. The threshold parameter $\kappa$ is the controlling factor of ABF for the length of the selected time window. As $\kappa$ increases, the length of the selected time window becomes larger. The value of 2 was chosen heuristically for this study as a compromise between selecting as many data points as possible and achieving the least data variability. Similar values of sensitivity controlling parameters in other data selection methods can be found (Thoning et al., 1989; Sirignano et al., 2010; Uglietti et al., 2011; Satar et al., 2016). In **Step 2**, values of 0.3 ppm and 0.5 indicate the threshold value for $s_i$ and $\pi_{missing}$. We denote them as $s_{i,threshold}$ and $\pi_{missing,threshold}$. It has been shown that less remote stations at lower altitudes require a larger value than 0.3 ppm because





of different mixing conditions. When performing ABF data selection at lower sites such as *HPB* and *SSL*, we recommend a higher $s_{i,threshold}$, such as 1.0 ppm instead of 0.3 ppm. However, throughout this study, we used the described parameter setting (0.3 ppm) for a methodical inter-comparison of selection methods at all stations. Potential influences of these parameter sizes ($s_{i,threshold}$ and $tr$) are discussed in Supplement S1.2 and S1.3.

**2.3 Other statistical data selection methods for comparison**

We compared ABF with three statistical data selection methods. The first method named SI is based on "steady intervals" (Lowe et al., 1979; Stephens et al., 2013). Steady intervals which are considered as baseline conditions are defined by a standard deviation being lower or equal than 0.3 ppm for 6 or more consecutive hours.

Secondly, we took a method applied by NOAA ESRL, which originated from Thoning et al. (1989). This selection routine
has been applied specifically for measurements of background $CO_2$ levels at Mauna Loa. This method (labelled as THO) was applied as described in the website (http://www.esrl.noaa.gov/gmd/ccgg/about/co2_measurements.html). The first step of THO examines the within-hour variability by selecting hours with hourly standard deviation less than 0.3 ppm. Secondly it computes hourly averages, and checks the hour-to-hour variability by retaining any two consecutive hourly values where the hour-to-hour difference is less than 0.25 ppm. The last step is based on the diurnal pattern (similar to ABF), by excluding
data from 11 a.m. to 7 p.m. LT due to potential influences of local photosynthesis.

The last method compared is a moving average technique (MA). A moving time window of 30 days and a threshold criterion of two standard deviations from the moving averages were applied to discard outliers. Afterwards, new moving averages and new threshold criteria were calculated for data exclusion. This step is repeated until no more outliers can be found. A more detailed description can be found in Satar et al. (2016).

**2.4 Seasonal-trend decomposition STL**

To analyze and compare the selected results from different data selection methods as well as the original validated data sets, we applied the seasonal-trend decomposition technique based on locally weighted regression smoothing (Loess), named STL (Cleveland, 1979; Cleveland et al., 1990). STL has been widely used on atmospheric $CO_2$ and other trace gases measurements (Cleveland et al., 1983; Carslaw, 2005; Brailsford et al., 2012; Hernández-Paniagua et al., 2015; Pickers and
Manning, 2015). It decomposes a time series of interest into a trend component $T$, a seasonal component $S$ and a remainder component $R$, which allows separately detailed analyses and comparisons of trend and seasonality. Two recursive procedures are included in the STL technique: an inner loop where seasonal and trend smoothing based on Loess are performed and updated in each pass, and an outer loop which computes the robustness weights to reduce the influences of extreme values for the next run of the inner loop (Cleveland et al., 1990).

For this study, we used the implemented function `stl` in R (R Core Team, 2017). Due to limitations of function `stl`, full time coverage of monthly data is needed in order to reduce the risk of large time gaps or unequal spacing (Pickers and





Manning, 2015). All data results were firstly aggregated to monthly averages. Then missing data were substituted by linear interpolation, using R function `na.approx` (Zeileis and Grothendieck, 2005). For the application of STL, two parameters need to be specified, which are the seasonal smoothing parameter $n_{(s)}$ ($s.window$ in function `stl`) and the trend smoothing parameter $n_{(t)}$ ($t.window$ in function `stl`). As $n_{(s)}$ and $n_{(t)}$ increase, the seasonal and trend components get smoother

(Cleveland et al., 1990). For an optimum compatibility in this study, the same parameters were chosen for all stations as $n_{(s)} = 7$ and $n_{(t)} = 23$, based on recommendation of Cleveland et al. (1990). Another parameters combination of $n_{(s)} = 5$ and $n_{(t)} = 25$ was also tested according to Pickers and Manning (2015) but with no significant differences in results.

## 3 Results and discussion

### 3.1 *Start time window*

ABF was applied to the validated hourly averages from all six stations with the parameter settings as described above. The detrended mean diurnal cycles were obtained with the *start time window* for each station by *starting selection* (see Fig. 2, for conventional mean diurnal plots see Supplement S2). The observed differences in the *start time windows*, as well as in the widths of the confidence intervals (grey shades in Fig. 2) can be explained by different site environments and thus differing data variabilities. The first subplot column (*HPB50*, *HPB93* and *HPB131*), represents the three sampling heights at *HPB*,

shows similar detrended diurnal patterns with similar *start time windows*. The decreasing amplitude with increasing sampling height indicates that the higher the sampling inlet is above the ground, the less affected it is by the local surface fluxes. The three *start time windows* suggest that the most stable period at *HPB* occurs during the last few hours of a day and also including midnight. However, in contrast to all other stations covering at least a full year, *HPB* data are only from September of 2015 to June of 2016. The results may be not fully comparable.

Regarding the second subplot column (*SSL*, *SNB* and *IZO*), the *start time windows* can be found from midnight on or later in the morning. The *start time window* for *SSL* encompasses its diurnal maximum, indicating that data variability is considerably smaller in the early morning than in the afternoon because of its vicinity to the Black Forest region, which has strong influence due to local photosynthetic activities (Schmidt et al., 2003). A similar diurnal pattern can be found at *SNB*. The influence of $CO_2$ sources is not as prominent as the effect of distant $CO_2$ sinks, since it is situated at the single summit

peak of Hoher Sonnblick only surrounded by mountains and glaciers with a negligible small number of tourists, thus anthropogenic activities are minimal. *IZO* is a special case, since it is located on a remote mountain plateau on the Island of Tenerife above the strong subtropical temperature inversion layer. Even though the *start time window* is limited to six hours, *IZO* presents a very ideal mean diurnal cycle for data selection from a potentially much longer time window.

On the right subplot column, both *ZSF* and *JFJ* find their *start time windows* around midnight (with more hours after

midnight). *ZSF* shows higher diurnal $CO_2$ amplitude compared with *JFJ*, but both sites show similar diurnal patterns. For the





choice of the *start time window* from the mean diurnal variation, relatively close or even local anthropogenic sources may influence the $CO_2$ at these two stations, possibly due to touristic influences.

## 3.2 Selection percentage

With the determined *start time windows*, ABF selected the data for all stations (see Fig. 3). And we calculated the

percentages of ABF selected data in all data for all stations, which are listed in the first column of Table 2. Among all stations, the highest percentage of data, found by ABF data selection belongs to *IZO*, with 36.2%. The following sites with intermediate percentages are stations *JFJ* (22.1%), *SNB* (19.3%) and *ZSF* (14.8%). For the three sampling heights at *HPB*, only 3.2% (50 m), 4.8% (93 m) and 6.2% (131 m) data were selected by ABF. At last, a similar percentage is found for *SSL* (3.8%), probably due to its higher data variability. To examine the characteristic growth of ABF selection percentages during

the selection process, we additionally calculated selection percentages after every major step. The detailed percentage results were listed in the Supplement S3.1. All the results of percentages show similar order of stations as above, and the selection percentages increase steadily step by step for all stations.

Since the stations were listed according to their altitudes, it was visible that all four selection percentages increase with altitudes, which indicated that measurements at higher altitudes could capture progressively well-mixed and hence

representative air. Therefore, linear least-squares regression was applied between the absolute altitude and the selection percentage for continental stations. *IZO* was on a remote island and therefore not comparable. As a result, a significant positive linear trend was observed (see coefficient in Table 2). The related figure of linear regression can be found in Supplement S3.2.

The selection percentages of ABF were again compared with those of the already mentioned statistical data selection

methods SI, THO and MA in Table 2, with the corresponding figure shown in Supplement S3.3. Since the selection percentages indicate not only the amount of data declared as representative but also show the characteristics of the selection methods, this criterion was used for further assessment. All other methods except for MA resulted in higher selection percentages for higher elevated stations (*IZO*, *ZSF*, *SNB* and *JFJ*) than lower elevated ones (*HPB* and *SSL*). ABF always performs the strictest in all cases. Based on the stepwise study of the selection percentages (see Supplement S3.1), the reason

for such low percentages is due to precise definition of the *start time window*. With *adaptive selection*, the selection percentages have grown but still maintain the lowest compared with other methods. SI and THO, on the other hand, show differences between high and low elevated stations. Compared with SI, THO is higher in low elevated stations, but lower in high elevated stations. A major limitation of SI seems to be the requirement of consecutive hours, in our case of six hours with 0.3 ppm standard deviation threshold, which might be too restrictive for low elevated stations. However, this criterion

results in a fairly large percentage for high elevated stations. At *ZSF*, *SNB* and *JFJ*, it results in the second largest, and even the largest in the case of *IZO*.

The highest selection percentages of approximately 80% are obtained with MA. But due to the minimal data variability of $CO_2$ measurements at *IZO*, the selection interval in MA becomes so small that the selection percentage becomes





considerably smaller compared with all other stations. However, *IZO* gets the largest selection percentages from all other selection methods. Thus, we conclude that MA does not work properly in the case of very well-mixed air (*IZO*). At all other stations, it is possible that MA declares too many data as representative. Therefore, MA was excluded from further analyses.

### 3.3 STL decomposed results

STL was applied to the validated data sets before and after selection with SI, THO and ABF, except for *HPB* due to its limited length in time, which is less than one year. Depending on data availability, STL was performed on $CO_2$ data from 2010 to 2014 at *SSL* and from 2012 to 2015 at *SNB*, while data inputs at *IZO*, *ZSF* and *JFJ* cover the complete period from 2010 to 2015. Figure 4 gives an overview of the decomposition in each component by STL.

### 3.3.1 Trend component

From the trend component the mean annual growth rate is estimated by linear regression (see Table 3). Based on the 95% confidence interval for the slope, a positive trend can be observed. Due to the overlapping of the confidence intervals, differences among VAL and all selected data sets at the same station are negligible. This indicates that the trend component is not influenced by the statistical data selection method, which agrees well with Parrish et al. (2012) for the study of baseline ozone concentrations that no significant differences of the long-term changes were found between the baseline and

unfiltered data sets. However, there is a tendency observed for all stations except for *SSL*. Compared to unselected data (VAL), the mean annual growth rates from selected data sets are systematically higher, in direction to the concentration levels at *IZO*, which here is considered as *reference* for better approximation of lower free tropospheric conditions. The deviation from this tendency at *SSL* probably is caused by stronger local influences as result of its lower elevation.

### 3.3.2 Seasonal component

The resulting seasonal components show systematic differences between VAL and selected data sets. The mean monthly variations were calculated on a monthly scale over the entire period from the analyzed data. Figure 5 (a) and (b) present the results at stations *ZSF* and *IZO*. At most stations (except for *IZO*) the seasonal amplitudes have been substantially reduced compared to VAL (see also Fig. 4). At *ZSF*, the averaged peak-to-peak seasonal amplitude, defined as mean seasonal maximum minus seasonal minimum, drops the most by 18.88% from VAL with the ABF selected data set. An explanation of

this reduction is $CO_2$ signal exclusion from local sources and sinks by data selection. When taking a closer look on monthly averages, lower $CO_2$ values are found in the selected data sets in the winter months from October to April, indicating that the $CO_2$ level is overestimated by VAL because of more dominant anthropogenic activities and almost no active vegetation. Higher values in the summer months from May to September explain an underestimation of VAL due to intensified vegetation signals. Similar patterns can be found at stations *SSL*, *SNB* and *JFJ* (see Supplement S4). *IZO*, as expected by its

location, shows always the smallest seasonal amplitude and nearly uninfluenced monthly results between VAL and selected data sets.





A time delay of one month in the mean seasonal maximum is found in Fig. 5 (a) at *ZSF* with selected data sets by SI and ABF (March) compared with the maximum from validated data (February). In addition, similar time shift can also be found by other selection methods at stations *SSL* (one month delay from February to March by SI and ABF) and *JFJ* (two months delay from February to April by SI, THO and ABF). As for stations *IZO* (April) in Fig. 5 (b) and *SNB* (March), the seasonal

maxima stay the same. Regarding the seasonal minima, no changes are observed, which is taken as an indicator for enhanced thickness of the mixing layer. Therefore, a better agreement of the seasonal peaks has been reached after data selection among all stations. This better represents the seasonal cycles of the baseline conditions.

### 3.3.3 Remainder component

The remainder component contains data with external and random influences. It has characteristics of random noise, being

basically different from site to site and statistically uncorrelated with the general signal of $CO_2$ concentrations in the lower free troposphere (Thoning et al., 1989). The standard deviation of the remainder component is taken here as a measure for external influences (see Fig. 4). Table 4 shows the calculated standard deviations from the remainder components at each station. Comparable results are derived from all selected data sets. *SSL*, as the lowest elevated station, exhibits the most variation. *IZO* with the least standard deviations proves to be the station least influenced by its surrounding environment.

The three alpine measuring stations (*ZSF*, *SNB* and *JFJ*) show similar intermediate results. From this perspective, STL performs well to show the site characteristics. Consequently, the noise of the remainder components, given in Table 4 decreases with increasing altitude of the continental mountain stations, which is in inverse relation to the selection percentage (Table 2). *IZO* was excluded in both regressions against altitude because of its maritime character. Further analyses with the remainder components could also yield the local influences and characteristics at each station in more

detail.

### 3.4 Correlation analysis

As mentioned above, data selection is defined here as approach of extracting a group of data to be the best representative for the lower free troposphere. Consequently, the selected $CO_2$ data sets from all stations should theoretically agree better among themselves. For validation, we took the combination of the trend and seasonal components from STL and examined the

correlations between each pair of stations (see the upper panel of Fig. 6). In general, most pairs show higher correlation coefficients with selected data from the different selection methods, especially between the three Alpine stations (*ZSF*, *SNB* and *JFJ*). This evaluation hence shows a similar result to the method presented by Sepúlveda et al. (2014) for identifying baseline conditions based on the correlation between distant measuring stations. Pairs including *IZO* after data selection by ABF show a noticeable increase in the correlation coefficients, meaning a better coherence between the reference station *IZO*

and the others. On the other hand, when selecting representative data more effectively, the results should contain less local and regional influences. Therefore, we compared the remainder components derived from STL pairwise to check whether the correlation coefficients decreased after data selection, as shown in the lower panel of Fig. 6. From all selection types



compared with VAL, ABF yields the maximum number of pairs of insignificant correlations. For the only two coefficients significant at 0.05 confidence level (*ZSF-SNB* and *ZSF-JFJ*), it drops largely from 0.75 to 0.48, and from 0.75 to 0.40 respectively, which cannot be observed by the other selection methods.

## 4. Conclusions and outlook

We presented a novel statistical data selection method, the Adaptive Baseline Finder (ABF), for $CO_2$ measurements at elevated GAW mountain stations. For validation and assessment of the data selection procedure we applied the method to six $CO_2$ data sets from 2010 to 2016, measured at GAW stations. For mountain stations in the European Alps, ABF selected an increasing percentage of data with growing elevation which is reasonable due to the underlying atmospheric dynamics. Comparing ABF with three other statistical data selection methods, all methods yielded rather consistent characteristics

across different stations. Nevertheless, among all the methods, ABF is the most restrictive in terms of number of selected data in the overall data sets.

In addition, we applied the time series decomposition tool STL to all validated and selected data sets. All statistical data selection methods resulted in the same annual trend in terms of the 95% confidence interval from the validated data sets while the seasonal signal varied substantially due to the reduced and delayed influences of $CO_2$ sources and sinks. We also

presented additional assessment of the proposed ABF method compared with the other statistical data selection methods based on correlation analysis. Both, higher correlation coefficients of the trend and seasonal components by STL and lower coefficients of the remainder indicate a better performance of ABF than the other methods SI and THO.

In all, this paper showed evidence that data selection based on the herein presented statistical properties enables practically feasible adjustments to individual conditions at GAW altitude measurement stations in Central Europe. This is a basic

prerequisite for the methodical application to a larger number of different stations and an essential step towards generalization. ABF, as an automated method, was shown to be a good option for mountain stations.

Hence for future research it is of substantial interest to test, whether this presented concept also holds in other regions and on other continents. Thus future investigations will target on including further altitude stations and on the question, how ABF will work with other air components. Also the question how to include seaside stations into a systematic and practically

generalizable approach for selecting representative data at GAW stations will be of essential concern.

## Acknowledgements

This work is supported by the scholarship from China Scholarship Council (CSC) under the Grant CSC No. 201508080110. Thanks to the Helmholtz Research School on Mechanisms and Interactions of Climate Change in Mountain Regions (MICMoR) for the support. The $CO_2$ measurements at Zugspitze and Schauinsland are supported by the German

Environment Agency (UBA). Thanks to Markus Wallasch for providing $CO_2$ data measured at Schauinsland and Ralf





Sohmer for technical support. The $CO_2$ measurements at Hohenpeissenberg are conducted by the German Meteorological Service within the ICOS Atmospheric Station Network. The $CO_2$ measurements at Jungfraujoch are supported by the Swiss Federal Office for The Environment, ICOS-Switzerland and the International Foundation High Alpine Research Stations Jungfraujoch and Gornergrat. Martin Steinbacher acknowledges funding from the GAW Quality Assurance/Science Activity

5    Centre Switzerland (QA/SAC-CH) which is supported by MeteoSwiss and Empa. The Izaña (*IZO*) $CO_2$ measurements were performed within the Global Atmosphere Watch (GAW) Programme at the Izaña Atmospheric Research Center, financed by AEMET. Thanks to Wolfgang Spangl from Austrian Environment Agency (UBA-At) for providing $CO_2$ data measured at Sonnblick.



**Table 1: Information of measured $CO_2$ data sets at six GAW mountain stations.**

| Station (GAW ID) | Sampling elevation (a.s.l.) | Time period (yyyy.mm) | Data provider |
|---|---|---|---|
| Hohenpeissenberg (*HPB*) | 984/1027/1065 m | 2015.09-2016.06 | DWD |
| Schauinsland (*SSL*) | 1210 m | 2010.01-2014.12 | UBA-De |
| Izaña (*IZO*) | 2403 m | 2010.01-2015.12 | AEMET |
| Zugspitze-Schneefernerhaus (*ZSF*) | 2670 m | 2010.01-2015.12 | UBA-De |
| Sonnblick (*SNB*) | 3111 m | 2010.01-2015.12 | UBA-At |
| Jungfraujoch (*JFJ*) | 3580 m | 2010.01-2015.12 | Empa |



**Table 2: Selection percentages of selected data in all data by different data selection methods. The bottom shows the linear regression coefficients of stations (*HPB* is represented by *HPB50*; *IZO* is excluded) altitudes and the selection percentages at significance level of 0.05 (\*\*\*).**

| Station ID | ABF | SI | THO | MA |
|---|---|---|---|---|
| *HPB50* | 3.2 | 13.9 | 21.7 | 79.8 |
| *HPB93* | 4.8 | 18.5 | 25.0 | 79.4 |
| *HPB131* | 6.2 | 21.3 | 27.3 | 79.8 |
| *SSL* | 3.8 | 17.3 | 25.3 | 81.4 |
| *IZO* | 36.2 | 82.2 | 56.0 | 60.5 |
| *ZSF* | 14.8 | 47.1 | 40.8 | 79.0 |
| *SNB* | 19.3 | 58.7 | 44.2 | 76.9 |
| *JFJ* | 22.1 | 62.1 | 46.3 | 77.6 |
| Linear regression coefficient ($\gamma^2$) | 0.996\*\*\* | 0.991\*\*\* | 0.986\*\*\* | 0.751 |





**Table 3: Mean annual growth rates (ppm y$^{-1}$) with 95% confidence intervals from linear regression applied on the trend components by STL.**

| Station ID | VAL | SI | THO | ABF |
|---|---|---|---|---|
| *SSL* | $2.12 \pm 0.14$ | $1.72 \pm 0.05$ | $2.05 \pm 0.10$ | $1.85 \pm 0.10$ |
| *IZO* | $2.24 \pm 0.03$ | $2.26 \pm 0.02$ | $2.25 \pm 0.02$ | $2.25 \pm 0.02$ |
| *ZSF* | $2.13 \pm 0.08$ | $2.16 \pm 0.05$ | $2.17 \pm 0.06$ | $2.19 \pm 0.06$ |
| *SNB* | $2.02 \pm 0.07$ | $2.06 \pm 0.06$ | $2.06 \pm 0.06$ | $2.08 \pm 0.04$ |
| *JFJ* | $2.13 \pm 0.03$ | $2.15 \pm 0.02$ | $2.14 \pm 0.02$ | $2.14 \pm 0.02$ |





**Table 4: Standard deviation of the remainder components by STL.**

| Station ID | VAL | SI | THO | ABF |
|------------|------|------|------|------|
| *SSL* | 1.58 | 1.27 | 1.27 | 2.03 |
| *IZO* | 0.34 | 0.33 | 0.30 | 0.30 |
| *ZSF* | 0.89 | 0.75 | 0.72 | 0.73 |
| *SNB* | 0.66 | 0.56 | 0.55 | 0.70 |
| *JFJ* | 0.56 | 0.45 | 0.48 | 0.47 |





**Figure 1: Locations of six European elevated mountain stations included in this study. Codes in brackets stand for countries (DE – Germany; AT – Austria; CH – Switzerland; ES – Spain).**





**Figure 2: Detrended mean diurnal cycles of validated CO$_2$ data sets (black) with 95% confidence intervals (grey) from six GAW stations (hours in LT). Measurements at *HPB* are differentiated by the sampling heights (e.g. *HPB50* for 50m a.g.l.). The covered time periods (top text), resulting *start time windows* (middle text, also in light blue shades) and mean diurnal amplitudes (bottom text) are shown in each subplot.**

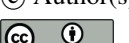

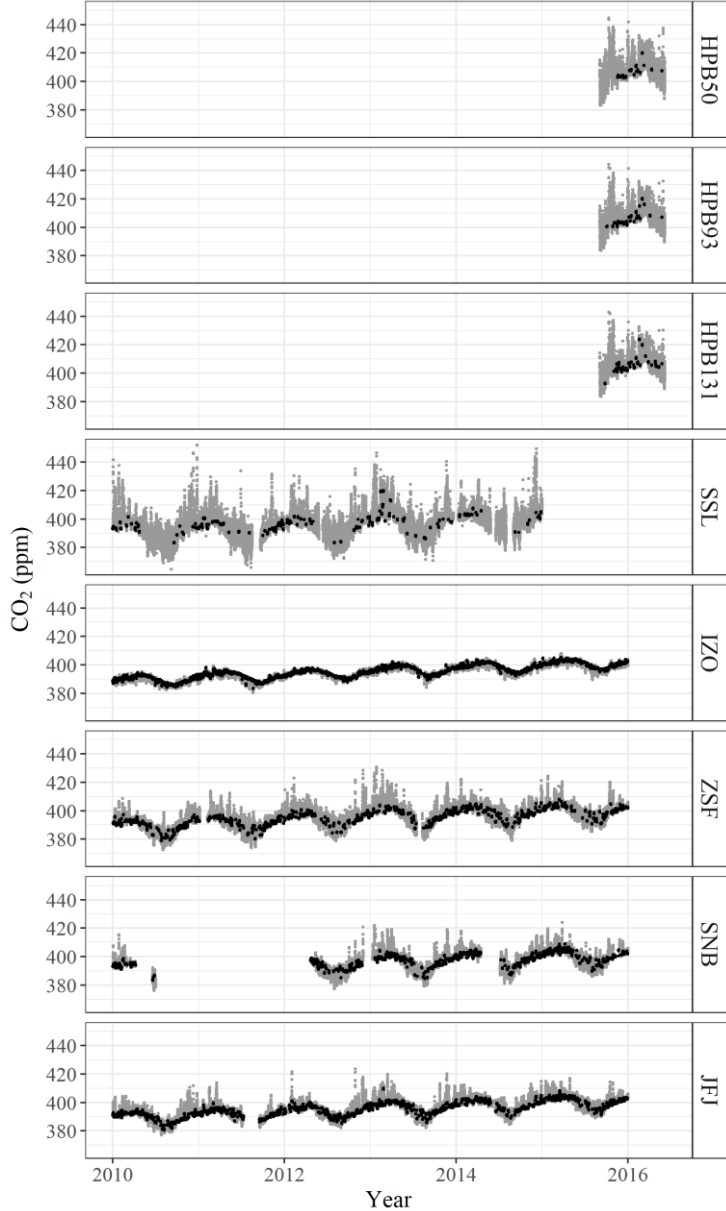

**Figure 3: Time series plots of validated CO$_2$ data sets (grey), and selected data sets by ABF (black) at six GAW stations.**





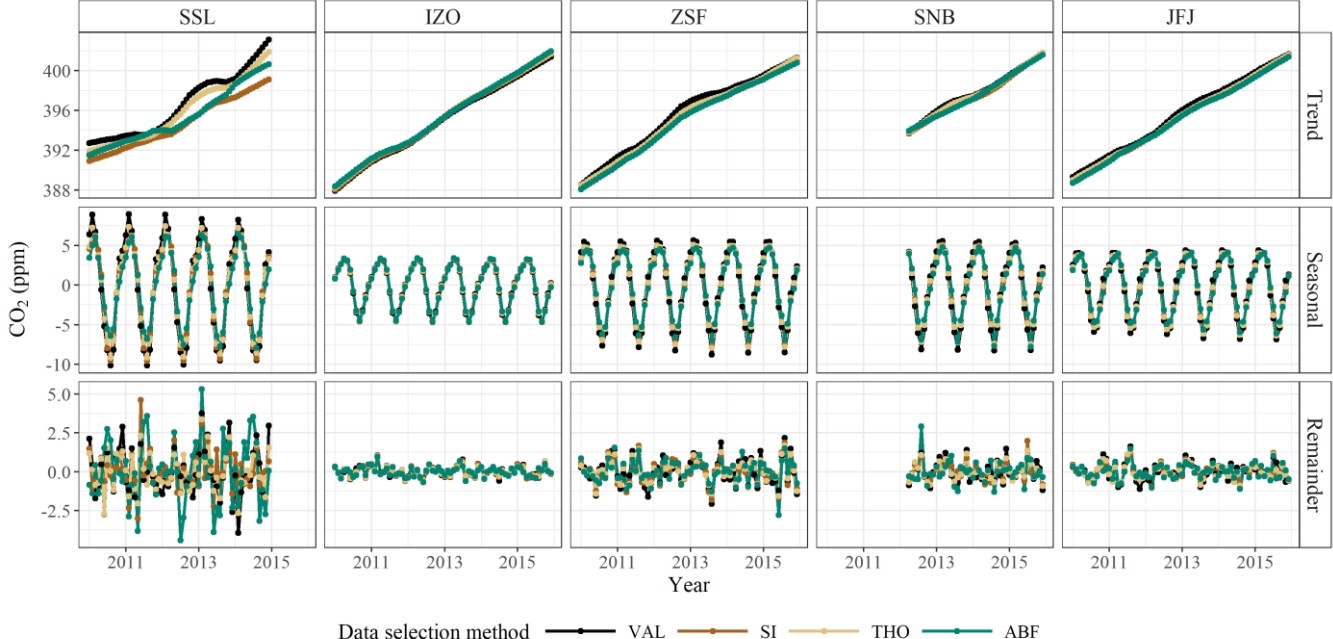

**Figure 4: STL decomposition results from VAL (black), SI-selected (brown), THO-selected (yellow) and ABF-selected (green) data sets at five GAW stations.**





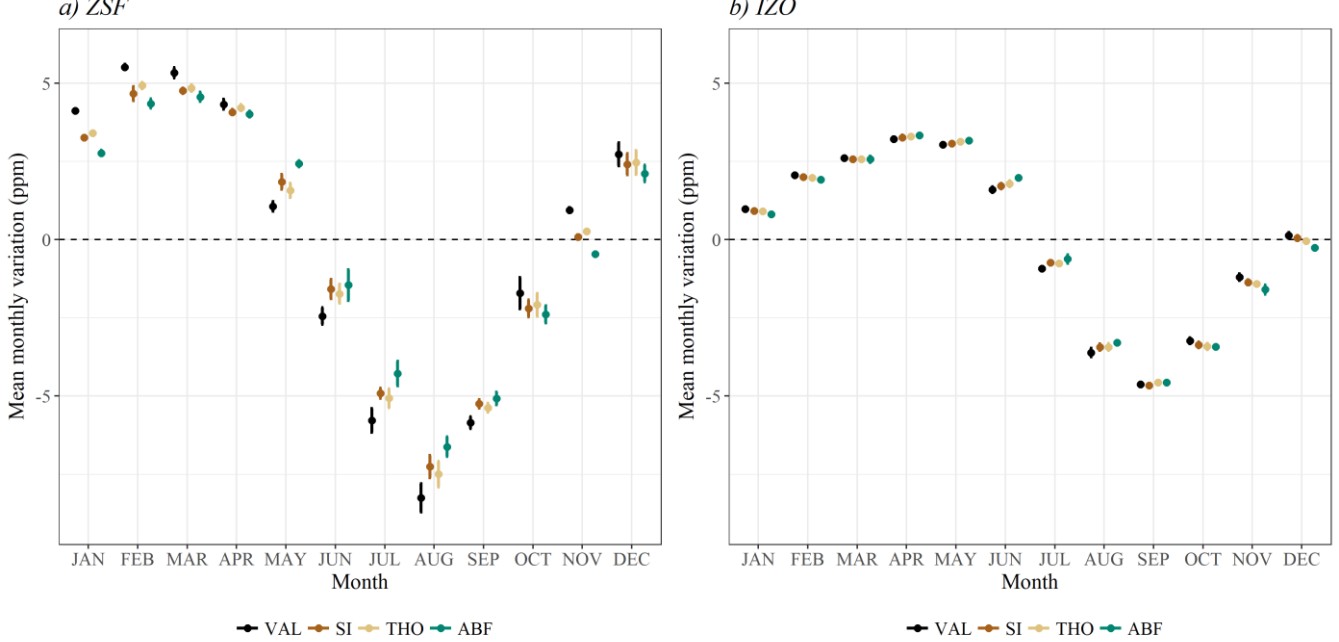

**Figure 5: Mean monthly variation of the seasonal component decomposed by STL at a) *ZSF* and b) *IZO* over the whole period. For better visualizing the results of selection methods, dots have been separated horizontally equidistant. The 95% confidence intervals are shown as error bars.**





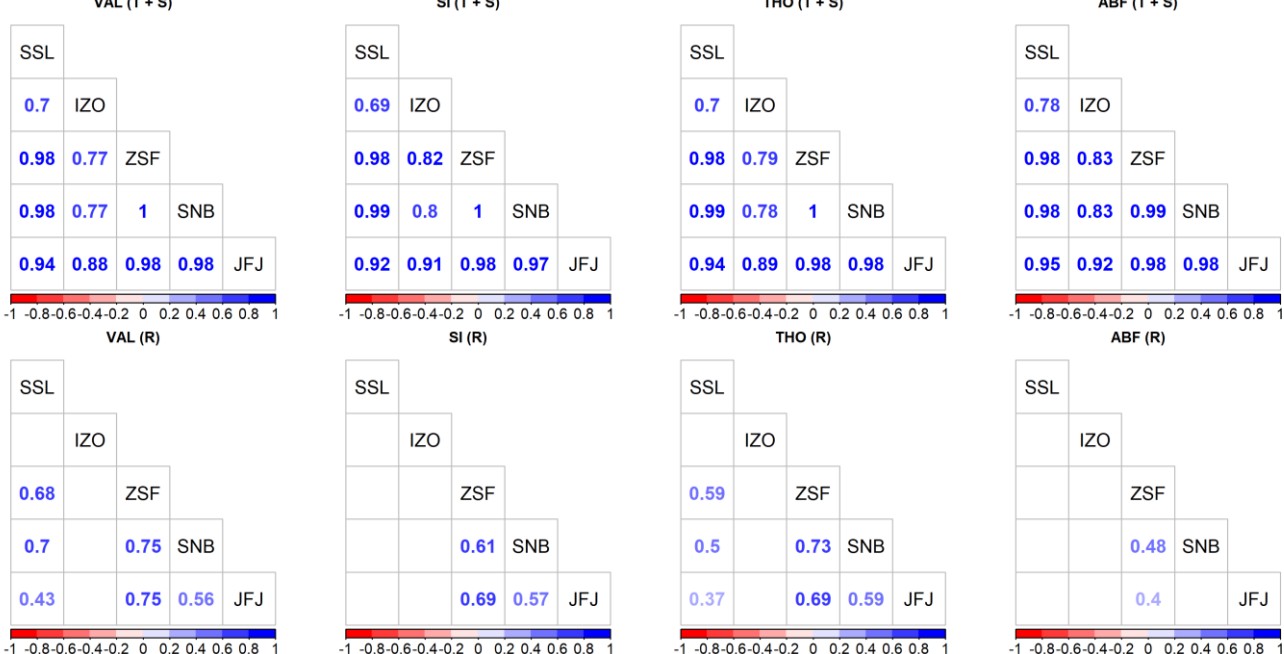

**Figure 6: Pearson correlation matrix of combinations of trend (T) and seasonal (S) components (upper panel), and only remainder (R) components (lower panel) at stations *SSL*, *IZO*, *ZSF*, *SNB* and *JFJ* by different selection methods as indicated on the top. Pearson correlation was applied due to normal distribution of data examined by Anderson-Darling test. Data used for correlation were chosen when available at all stations (2012.04 – 2014.12). The blue color scale reflects the strength of positive correlation. Correlations with no significant coefficient at 0.05 confidence level were left blank.**





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
