# Peer review of "Adaptive selection of diurnal minimum variation: a statistical strategy to obtain representative atmospheric CO2 data and its application to European elevated mountain stations"

_Atmospheric Measurement Techniques, 2017_

## Referee Comment (RC1) · Anonymous Referee #1 · 7 Oct 2017

Yuan et al. present a data selection method for records of atmospheric CO2 mole fraction observations from mountain locations. Their method, the adaptive baseline finder (ABF), is an interesting one and in that sense worth publishing. However, unfortunately the manuscript in its current form remains very descriptive and does not include clear conclusions on how the community would benefit from using this method in comparison to the other methods to which ABF is compared.

Another main point is that the English should be checked by a language editor, as in several places the manuscript is not written in correct English (e.g. articles are often

omitted and commas are used incorrectly).

All in all, I think the authors have done a substantial amount of interesting work, and could be worth publishing after taking into account the specific comments below and especially focus on placing their work in larger context and making more explicit what the use of ABF could contribute to the field.

Specific comments:

Page 1 line 21: 'measuring sites' should be replaced by 'measurement sites', throughout the manuscript.

Page 1 line 22: Why would this lead to a bias when comparing different stations? Only when the data of these different stations has been selected with different methods.

Page 1 line 23: pattern -> patterns

Page 1 line 24: 'measuring records' -> records of atmospheric CO2 observations

Page 1 line 27: implemented -> included/applied

Page 1 line 27: Among the studied methods, our ABF method . . .

Page 1 line 27: This is very descriptive: lower percentage of selected data; is this 'better'? What does it imply to have less or more data selected?

Page 1 line 30: STL is not explained

Page 2 line 13: what do you mean by correction for interference from other GHG?

Page 2 line 24: here it would be good to elaborate on the work of Uglietti et al. 2011 (ACP), which is referred to on the same page.

Page 2 line 29: explain why afternoon values should be excluded.

Page 3 line 3: MHD flasks are only sampled during restricted base line conditions, so no filtering is applied.

Page 3 line 6: Hawaii, USA. Also add Switzerland for JFJ.

Page 3 line 18: what is REBS?

Page 3 line 23: why do the authors choose to focus on mountain sites only? This should be made more clear in the manuscript.

Page 4 line 7-9: what do these classifications mean? E.g. "weakly influenced, constant deposition" is not very clear.

Page 4 line 13: did you use hourly data or higher time resolution? This is not clear from this section.

Page 4 line 15-16: specify which reference is for which station.

Page 4 line 19-20: This sentence is very vague, make more clear what the motivation of this research is.

Page 4 line 21-22: This sentence is not clear: what traffic activities are relevant to the mountain sites? And why is vegetation active in the afternoon only? How about respiration?

Page 4 line 22-24: this sentence is not clear. What do you mean by 'which in turn in an effective tool'? What tool?

Page 4 line 25: explain PBL and explain the changing degree of entrainment.

Page 4 line 27-31: The level of English needs to be assessed particularly in these sentences.

Page 5 line 7: What is the time resolution of the data sets?

Page 5 line 11-20, and page 6 line 6-22: Revise English especially here, including use of complete sentences including articles ('the') and correct use of commas.

Page 6 line 26: This is a vague sentence, data only exists on a single day, so why talking about selecting it in 'any day'?

Page 7 line 15: photosynthesis starts long before 11 a.m.

Page 8 line 10: Why hourly? How did you define hourly values? As the average of the whole hour? Or just last part? Is the hour defined at the beginning of the averaging interval or at the end? This is important information and should be included in methods.

Page 8 line 15: Does it make sense to have different windows at the different levels?

Page 8 line 19: The results ARE not fully comparable. Does it even make sense to analyze such a short record which does not even give a complete annual cycle?

Page 9 line 2: It would make sense to look at the differences by season, as the diurnal cycle is not the same throughout the year. Also, the data sets all cover different time periods, so it is difficult to compare.

Page 9 line 4-10: Revise English.

Page 9 line 6-9: But what do these percentages actually mean? This is too descriptive and needs more analysis and perspective.

Page 9 line 10: what is 'major step' and what do the percentages by each step mean?

Page 10 line 3: This previous section remains very descriptive. What do the differences between all methods mean, and what is more useful for what type of analysis? This needs more work.

Page 10 line 10: What is the use of comparing growth rates for different time periods? Growth rates are very variable from year to year, so choosing a different period gives different growth rates.

Page 10 line 11: A positive trend in what? In the $CO_2$ concentrations in general?

Page 10 line 12: Explain VAL

Page 10 line 12: what differences?

Page 10 line 15: What do you mean by tendency?

Page 10 line 18: 2015 had a much higher growth rate compared to the years before, so that also influences the results at SSL. Why not including 2015? It is publically available through ObsPack.

Page 10 section 3.3.1: I do not understand the added value of this paragraph. It should include more details on what was exactly studied and more conclusive remarks instead of only descriptive statements.

Page 10 line 20: this is not clearly described (the difference between val and selected data).

Page 10 line 24: this percentage is given too much precision.

Page 10 line 27: if VAL is all validated data it can never over- or underestimate CO2 levels, as they are the actual observations!

Page 11 line 1-7: Very descriptive, add more details and analyses and perspective.

Page 11 line 6: explain in more detail 'thickness of the mixing layer'.

Page 11 line 14: what does 'least standard deviations' mean?

Page 11 line 14: we already knew that IZO is least influenced.

Page 11 line 15: what are intermediate results?

Page 11 line 19: could, but why not done?

Page 11 line 25, figure 6: why is red included in the color scale as those values do not occur? Also the caption of figure 6 contains a lot of information that should be included in the main text as well (pearson corr. matrix etc).

Page 12 line 1: what does this mean/imply?

Page 12 conclusions: should be especially checked for level of English.

Page 12 line 7: not all 6 cover this period.

Page 12 line 7: rewrite, ABF does not select..

Page 12 line 8: growing elevation?

Page 12 line 10: but what does it mean/imply that is the most restrictive? When would you recommend the ABF method?

Page 12 line 14: what do reduced and delayed mean here?

Page 12 line 18-19: what do you mean?

Page 12 line 21: how applicable is the method to other stations?

Figure 1: add larger map to know which region of the world this is.
* * *

---

## Referee Comment (RC2) · J. Kim (Referee) · 8 Nov 2017

This work presents a new statistical algorithm, named ABF, for identifying "baseline" levels from CO2 measurements. The title of the work refers to elevated mountain sites as its application focus, but the work also includes some analysis of non-mountain sites as well. While there are some issues that I would like to see the authors address, overall I do feel the authors have done a good job of presenting a unique algorithm and comparing it to other frequently used methods in the measurement community, and as such I suggest that the manuscript be published with some revisions.

[Figure]

Before I proceed with my comments on the paper, I would like to comment on the term "baseline" itself. My concern is that the definition of "baseline" is very subjective open to interpretation. For example the authors mention that ABF in this study was used specifically to identify periods of free troposphere concentrations in the high elevation sites, and that is certainly one valid definition of "baseline". With this definition, however, sites that may have statistically stable concentrations at certain times of the day but do not necessarily measure the free troposphere will by definition have no "baseline". If the definition of "baseline" was "typical concentrations you would probably measure at a certain location at a certain time" with the goal of creating a global spatial map of average concentrations, I suppose you would end up with something close to the trend and seasonal components in the STL analysis, which you may (or may not) be able to find through statistical methods such as ABF. On the other end of the spectrum, for a regional modeler, the useful definition of "baseline" would be whatever concentrations enter the modeling domain and not necessarily any clean/stable condition, and if the air was polluted coming into the grid box then the model needs to know about it. I've seen attempts to distinguish between "baseline" and "background" to try to navigate through the subtle (and sometimes not-so-subtle) differences in definitions, but in my view all attempts at defining "baseline" is inherently subjective and the best practice is to be specific about what the particular definition for the study is, and that definition should encompass the specific intended use of this definition. All this to say, I feel the name Adaptive BASELINE Finder, while sounding nice, can be misleading. I would suggest that the authors consider another name, but will leave the decision to the authors.

[General comments, questions]

- P5, ln 15: Why the window of 6 hours? I suppose this assumes that baseline conditions occur for longer than 6 hours? Have you tried shorter windows and found you come to the same conclusion? I almost wonder whether it would be more beneficial as a general algorithm to have as short a window as possible, such that the window never exceeds the actual window of a baseline occurrence?

- P10, ln 15: The increase in the mean annual growth rates is within the noise, I'm not sure that much can be made of this.

- Figure 2: I have a hard time understanding this figure. First off, the figure seems to represent data from the full data set (spanning years), and yet the method describes that the baseline "window" is adaptive, potentially changing each day and by season. What criteria was used to derive a representative window for the whole period?

- P10, ln 27: Regarding "active vegetation", wouldn't signals from respiration also explain these results, and wouldn't that also be one form of active vegetation? I think this possibility can't be ignored since the authors suggest that the lower VAL values in summer are likely due to vegetation. Are the anthropogenic emission activities in this region such that you would expect emissions only in winter, or are they small enough to be masked by the summer drawdown? I do think that the authors' interpretations on the findings are likely to be correct, however I do think that a much deeper analysis of the data (perhaps beyond the scope of this paper) may be needed to conclusively determine the source of these discrepancies.

- One discussion I think is missing is regarding the "adaptiveness" of the algorithm, in other words do the results show baseline windows changing with season. The authors state this as a strength of the ABF (P4 ln 29), so I had expected this to be one of the early points of discussion.

[Minor comments]

- Page 4, ln 10: "At last", change to "Finally"?

- P4, ln 27: "No upwind air masses with depleted CO2 levels by photosynthesis of vegetation like in summer are recorded." -> "Unlike summer, no upwind air masses with depleted CO2 levels by photosynthesis of vegetation are recorded."

- P5, ln 12: "but preserves of the diurnal pattern." -> "while preserving the diurnal pattern."

- P6, ln 10: "Step 3" is not actually a step, but a general description of Step 5 and 6. Perhaps it makes more logical sense to include it in "Step 2", presenting it as an "If/Else" step.

- P9, ln 5, Table 2: Can the authors clarify whether the percentages are based on just the time windows considered in the algorithm or the complete dataset?
* * *

---

## Author Comment (AC1) · 5 Dec 2017

**Ye Yuan et al., Adaptive Baseline Finder, a statistical data selection strategy to identify atmospheric CO₂ baseline levels and its application to European elevated mountain stations**

Answers to Anonymous Referee #1

The referee's comments are in black, answers are in blue.

**Short notice:**

According to the suggestion from J. Kim (Referee #2), we changed the name of our method "Adaptive Baseline Finder (ABF)" into "Adaptive Diurnal Minimum Variation (ADMV)". All the names and abbreviations of this method have been adjusted throughout the answer.

Yuan et al. present a data selection method for records of atmospheric $CO_2$ mole fraction observations from mountain locations. Their method, the adaptive baseline finder (ABF), is an interesting one and in that sense worth publishing. However, unfortunately the manuscript in its current form remains very descriptive and does not include clear conclusions on how the community would benefit from using this method in comparison to the other methods to which ABF is compared.

Another main point is that the English should be checked by a language editor, as in several places the manuscript is not written in correct English (e.g. articles are often omitted and commas are used incorrectly).

All in all, I think the authors have done a substantial amount of interesting work, and could be worth publishing after taking into account the specific comments below and especially focus on placing their work in larger context and making more explicit what the use of ABF could contribute to the field.

We would like to thank the referee for the very detailed and constructive comments. Besides answering all the specific comments respectively, we also added more explanations and arguments mainly in the Results and discussion, as well as the Conclusion section.

**Specific comments:**

Page 1 line 21: 'measuring sites' should be replaced by 'measurement sites', throughout the manuscript.

This was corrected.

Page 1 line 22: Why would this lead to a bias when comparing different stations? Only when the data of these different stations has been selected with different methods.

We apologize for the misleading wording. We replaced "bias" by "reduced compatibility."

Also, it needs to be noted that different stations do have different methods for data selection and data processing due to different station characteristics and measurement conditions (including instrument, etc.). In a way, it is also true that data compatibility needs to be improved within GAW network. This methodological approach focuses on common structural features of measurement data from mountain stations with the aim of finding a more general solution for the selection of representative measurement data. The aim is to improve the compatibility of the data and to facilitate the conclusion from a point measurement to a larger area.

Page 1 line 23: pattern -> patterns
    This was corrected.

Page 1 line 24: 'measuring records' -> records of atmospheric CO2 observations
    This was corrected.

Page 1 line 27: implemented -> included/applied
    This was corrected.

Page 1 line 27: Among the studied methods, our ABF method …
    This was corrected, and the full name was added in the text above (because of the abbreviation used here).

Page 1 line 27: This is very descriptive: lower percentage of selected data; is this 'better'? What does it imply to have less or more data selected?
    We added by the end of this sentence, "…, which can be understood as a better representation of the lower free troposphere."

Page 1 line 30: STL is not explained
    We rewrote the sentence, "The measured time series were analyzed for long-term trend and seasonality by seasonal-trend decomposition technique."
    And STL would be explained later in detail in Section 2.4.

Page 2 line 13: what do you mean by correction for interference from other GHG?
    We added an example for the potential interference, "such as water vapor."

Page 2 line 24: here it would be good to elaborate on the work of Uglietti et al. 2011 (ACP), which is referred to on the same page.
    Since the method for calculating the background corrected $CO_2$ record was the same as Satar et al. (2016), we included the citation of Uglietti et al. (2011) at the end of Section 2.3 for the method description.
    Regarding the use of the transport model, we elaborated on the reference to the work of Uglietti et al. (2011) in the same paragraph but some lines further below when the modeling techniques were mentioned. The added sentence is, "Uglietti et al. (2011) focused on the origins of atmospheric $CO_2$ at Jungfraujoch (Switzerland) by the FLEXible PARTicle dispersion model (FLEXPART)."

Page 2 line 29: explain why afternoon values should be excluded.
    We added, "…due to the influence of convective upward transport."

Page 3 line 3: MHD flasks are only sampled during restricted base line conditions, so no filtering is applied.
    We apologize for the misleading wording. It is correct that MHD flasks were only sampled during Restricted Baseline Conditions (RBC), i.e. during periods with specific wind directions and

wind velocities requirements.

When citing this paper, we wanted to stress the approaches for aiming on a regional analysis instead of filtering technique. More details can be seen in Section 3 (Data selection) of Sirignano et al. (2010).

For this sentence, we rephrased as, "Threshold limits of 300 ppb for CO and 2000 ppb for $CH_4$ were defined by Sirignano et al. (2010) to perform a regional analysis of $CO_2$ data at Lutjewad (the Netherlands) and Mace Head (Ireland)."

Page 3 line 6: Hawaii, USA. Also add Switzerland for JFJ.

These were added.

Page 3 line 18: what is REBS?

We rephrased the sentence as, "Ruckstuhl et al. (2012) developed a method based on robust local regression, called Robust Extraction of Baseline Signal, to estimate…"

Page 3 line 23: why do the authors choose to focus on mountain sites only? This should be made more clear in the manuscript.

This work concentrates on finding common properties for the lower free troposphere in ground-based measurements. As the first approach data have been taken from mountain stations due to their remote location with limited anthropogenic influence and increased representativeness. We also focus on mountain sites only because mountain stations have a diurnal variation which results in a daily time window for well-mixed air with a better representation of the lower free troposphere. The explanation in Section 2.2 was improved considerably, which can be seen in the answers below.

Also, we added at the beginning of Section 2.1, "The data have been taken from mountain stations due to their remote location with limited anthropogenic influence and increased representativeness."

Page 4 line 7-9: what do these classifications mean? E.g. "weakly influenced, constant deposition" is not very clear.

We added the explanation at the beginning of this sentence, "Henne et al. (2010) presented a method of categorizing site representativeness based on the influence and variability of population and deposition by the surface fluxes."

Page 4 line 13: did you use hourly data or higher time resolution? This is not clear from this section.

Both time resolutions hourly and half-hourly are available. We used hourly data throughout the work except for the evaluation of the influences of different time resolutions (see Supplement S1.3).

Therefore, we added, "For this study, hourly data were used consistently, unless otherwise indicated."

Page 4 line 15-16: specify which reference is for which station.

We added the station names in the sentence, "Schmidt et al. (2003) for *SSL*, Gilge et al. (2010)

for *HPB* and *SNB*, Gomez-Pelaez et al. (2010) for *IZO*, Risius et al. (2015) for *ZSF* and Schibig et al. (2015) for *JFJ*."

Page 4 line 19-20: This sentence is very vague, make more clear what the motivation of this research is.

We rephrased this sentence as, "ADMV is a tool for automatic and systematic analysis of diurnal $CO_2$ cycles at elevated mountain stations in order to select consecutive time sequences with minimum variation, which can be regarded as representing well-mixed air conditions."

Page 4 line 21-22: This sentence is not clear: what traffic activities are relevant to the mountain sites? And why is vegetation active in the afternoon only? How about respiration?

We rephrased this sentence as, "For example, at ZSF, these can be characterized by anthropogenic $CO_2$ sources, detectable especially in winter during the day, whereas in summer the convective upwind transport results in a strong impact of air masses with depleted $CO_2$ concentrations due to photosynthesis at lower altitudes. Plant respiration activities, which may contribute small amounts, are primarily not visible in the convective upwind air masses (which arrive at mountain sites predominantly in the afternoon). Although high elevated mountain stations do not have vegetation in their surroundings, mountain stations at lower altitudes but still in the vegetation zone may be influenced by plant respiration, especially at night."

Page 4 line 22-24: this sentence is not clear. What do you mean by 'which in turn in an effective tool'? What tool?

We apologize for the misleading wording. We rephrased "is an effective tool for data selection" as, "can be used for selecting representative data."

Page 4 line 25: explain PBL and explain the changing degree of entrainment.

PBL was introduced in Section 2.1 (Page 4 line 1), as "planetary boundary layer."

And hereby we rephrased "(e.g., due to changing degree of entrainment of PBL air)" as "because of variations in the dynamics of transport to the site (e.g., Birmili et al., 2009; Herrmann et al., 2015)."

Page 4 line 27-31: The level of English needs to be assessed particularly in these sentences.

We rephrased this part as, "…whereas in winter, significantly longer stable periods occur. In winter, no upwind air masses with depleted $CO_2$ levels due to photosynthesis of vegetation are recorded. To receive as much representative data as possible, it is desirable to select the time window dynamically. ADMV is constructed to select a subset from the measured data, being best representative for baseline conditions with an adaptive selected time window specific for every day."

Page 5 line 7: What is the time resolution of the data sets?

The time resolution is hourly. Therefore, we added "hourly" in the sentence.

Page 5 line 11-20, and page 6 line 6-22: Revise English especially here, including use of complete sentences including articles ('the') and correct use of commas.

An English proofreading has been done throughout the paper.

This part is shown in the following.

"**Step 1**: Detrending is done by subtracting a 3-day average for each day, including the neighboring two days. It is the shortest possible time window to remove sudden changes in the time series related to the previous and posterior days while preserving the diurnal pattern.

**Step 2**: The overall mean diurnal variation, $\bar{d}_i$ (i = 0 to 23 h), is calculated from the complete set of detrended data.

**Step 3**: The standard deviations $s_{\Delta_j}$ from the overall mean diurnal variation $\bar{d}_i$ are calculated on a moving window $\Delta_j$ ($j = 6\,h$). To be able to place a full set of 24 moving time windows over the overall mean diurnal variation, time windows across midnight (e.g., 6 h from 11 p.m. to 4 a.m. LT) are also included, that is, its first $j$ hours are appended to the end of the 24 h in the overall mean diurnal variation. The time window with the smallest standard deviation is selected as the *start time window*.

**Result**: The *start time window* $[i_{start}, …, i_{end}]$."

Page 6 line 26: This is a vague sentence, data only exists on a single day, so why talking about selecting it in 'any day'?

We rephrased the sentence as, "We always label the data as "selected" once it has been selected by ADMV."

Page 7 line 15: photosynthesis starts long before 11 a.m.

We rephrased "potential influences of local photosynthesis" as "transported air influenced by photosynthesis."

Page 8 line 10: Why hourly? How did you define hourly values? As the average of the whole hour? Or just last part? Is the hour defined at the beginning of the averaging interval or at the end? This is important information and should be included in methods.

Hourly values are used because of the availability of hourly averages as the highest time resolution in the World Data Center for Greenhouse Gases (WDCGG). Therefore in order to keep the format of input data constant for ADMV method, we calculated the average of the whole hour for all data sets. The time stamp for the hourly average was defined as the beginning of the averaging interval.

Moreover, originally ADMV was developed based on 30-min time resolution at the station ZSF. Therefore ADMV method can also handle data with higher time resolution than one hour.

We added in Section 2.1, "In addition, the time stamp was defined at the beginning of the averaging interval."

Page 8 line 15: Does it make sense to have different windows at the different levels?

The different *start time windows* at the different levels result automatically from the ADMV method. It always searches for the optimal *start time window* based on specific data sets. In our opinion, these are very interesting and valuable results, which reflect to some extent the different characteristics of different measurement sites and also different levels. In this respect, the different time windows at the different sampling levels are results of differences in the dynamics of atmospheric transport.

Page 8 line 19: The results ARE not fully comparable. Does it even make sense to analyze such a short record which does not even give a complete annual cycle?

We agree that the data were not fully comparable because the time period was too short in contrast to the other stations. However, the results showed that for time periods shorter than a full year, the ADMV method was still applicable to the data from the tower measurements, which highlights the flexibility of the approach.

Page 9 line 2: It would make sense to look at the differences by season, as the diurnal cycle is not the same throughout the year. Also, the data sets all cover different time periods, so it is difficult to compare.

We agree that there are differences in diurnal patterns among seasons. We also applied the ADMV method separated by season, i.e., data sets were processed and selected by the ADMV method only during a specific season over the whole time period. However, we found that the *start time windows* didn't differ significantly (see Supplement S1.1).

Regarding different time periods of the sites we also included data of 2015 for *SSL*. Now except for *HPB*, all the measurement sites cover the same time period.

Page 9 line 4-10: Revise English.

The English proofreading has been done throughout the paper.

And we rephrased this paragraph as,

"With the determined *start time windows*, ADMV selected the data for all stations (see Fig. 3). In addition, we calculated the percentages of ADMV selected data values among all values of the complete datasets for all stations, which are listed in the first column of Table 2. The higher the selection percentage is the more well-mixed air is measured at the station, which is assumed to be a representation of lower free tropospheric conditions. This holds especially for *IZO*. Because of this the greatest amount of accepted data points with 36.2% was found at this station. The sites with intermediate percentages are *JFJ* (22.1%), *SNB* (19.3%), and *ZSF* (14.8%). For the three sampling heights at *HPB*, only 3.2% (50 m), 4.8% (93 m), and 6.2% (131 m) of the data were selected by ADMV. Finally, a similar percentage was found for *SSL* (4.0%), probably due to its higher data variability."

Page 9 line 6-9: But what do these percentages actually mean? This is too descriptive and needs more analysis and perspective.

We added as mentioned above, "The higher the selection percentage is the more well-mixed air is measured at the station, which is assumed to be a representation of lower free tropospheric conditions. This holds especially for *IZO*. Because of this the greatest amount of accepted data points with 36.2% was found at this station."

Page 9 line 10: what is 'major step' and what do the percentages by each step mean?

These different definitions of selection percentages were explained in the Supplement S3.1.

Besides, we rephrased the sentence as, "we additionally calculated selection percentages after completing both the *starting selection* and *adaptive selection* steps mentioned in Section 2.2 (see Supplement S3.1)."

Page 10 line 3: This previous section remains very descriptive. What do the differences between all methods mean, and what is more useful for what type of analysis? This needs more work.

We explained the differences of all methods and the potential reasons in the paragraph above (please see page 9 line 24–33). For ADMV method, the detailed stepwise results of selection percentages were made in Supplement S3.1. For SI and THO methods the major difference is the requirement of consecutive hours. As for MA method, the selection criteria would become too strict for stations with very small data variability (e.g., *IZO*).

On the other hand, this paper focuses more on the mechanism and results of ADMV method. Our intention is to give a clear and detailed instruction on this data selection method and provide options to the users. The advantage of ADMV method can be seen in the Conclusion section. To compare more thoroughly of different data selection methods and present a clear strategy of applying different methods require further researches and are beyond the focus of this paper.

Page 10 line 10: What is the use of comparing growth rates for different time periods? Growth rates are very variable from year to year, so choosing a different period gives different growth rates.

The STL technique has been re-run. The underlying time period is 2010 to 2015 for all sites except for SNB, for which data of 2010 to 2011 are missing. We changed the color for SNB to gray in Table 3 and Table 4 for differentiation.

By comparing growth rates, we aim at showing there are no significant influences on the trend components by these data selection methods.

Page 10 line 11: A positive trend in what? In the CO2 concentrations in general?

We rephrased as, "Based on the 95% confidence interval for the slope, a positive trend i.e. increasing $CO_2$ concentrations are observed."

Page 10 line 12: Explain VAL

VAL is validated data and thus delivered to the GAW data bases (Level 2 data). It has been explained in Section 1 (Page 2 line 12).

Page 10 line 12: what differences?

We rephrased as, "differences in the mean annual growth rates…"

Page 10 line 15: What do you mean by tendency?

What we can observe from the resulted mean annual growth rates is that, the growth rates resulted after data selection are mostly higher than the ones of validated datasets and always approaching toward the values at station *IZO*. However, this is not statistically significant based on the confidence intervals. That is the reason we meant by tendency.

We rephrased as, "Moreover, the following fact is observed for all sites except for *SSL*."

Page 10 line 18: 2015 had a much higher growth rate compared to the years before, so that also influences the results at SSL. Why not including 2015? It is publically available through ObsPack.

Thank you very much for your suggestion. Now the $CO_2$ data of 2015 for *SSL* are included

and all respective analyses have been re-run.

Page 10 section 3.3.1: I do not understand the added value of this paragraph. It should include more details on what was exactly studied and more conclusive remarks instead of only descriptive statements.

This paragraph showed the results of the trend components of all data sets at all stations. The comparison of trends based on datasets with and without data selection methods clearly indicated that mean annual growth rates were not significantly different.

To include more details, we added the following arguments in the text.
"Compared to unselected data (VAL), the mean annual growth rates based on selected data sets are systematically higher approaching the growth rates at *IZO*. *IZO* can be considered as better representing the lower free tropospheric conditions and agrees well with the mean annual absolute increase during last 10 years (2.21 ppm yr$^{-1}$) reported by WMO (2017a). The exception at SSL is probably is caused by stronger local influences as a result of its lower elevation. Besides, the confidence intervals of the mean annual growth rates are always smaller after data selection, which improves the precision of trends."

Besides, we also added a general description at the beginning of the section, "The following sections discuss the resulting components obtained by STL, namely the trend component over the observation period, the seasonal component and finally the remainder component."

Page 10 line 20: this is not clearly described (the difference between val and selected data).

As mentioned above, VAL was validated data and thus delivered to the GAW data bases, which was taken as data input for our study. Selected data were resulted data after different data selection methods.

We rephrased, "systematic differences which calculated for validated (VAL) and selected data sets…"

Page 10 line 24: this percentage is given too much precision.

We changed it to "18.9%."

Page 10 line 27: if VAL is all validated data it can never over- or underestimate CO2 levels, as they are the actual observations!

We apologize for the misleading wording.

VAL data are validated correct measurements, adjusted to the international standard reference scales and following the Global Atmosphere Watch quality objectives. Nevertheless, due to the different time scales of transportation effects VAL data may contain values from a time period where the well mixing assumption is violated (short time events). Since we referred VAL as validated unselected measurements, the $CO_2$ levels mentioned here refer to the background level of $CO_2$ which are supposed to take place at the measurement sites.

If all validated data are used, this would result in an overestimation of the atmospheric $CO_2$, due to the dominance of anthropogenic activities and no active vegetation in winter. Thus, it indicates that the VAL data are not representative.

Page 11 line 1-7: Very descriptive, add more details and analyses and perspective.

We rewrote this paragraph partly and added more content as the following.

"The magnitude of these delays may be related to mixing features in the lower free troposphere. Rapid changes are usually observed close to sources and sinks, e.g., from anthropogenic and biogenic activities. Thus, the higher the station is above the boundary layer, the later the maxima during the winter can be observed, because of the late response due to inhibited mixing conditions. However, this delay does not occur for the minima during the summer because of the very effective upward transport and more favorable mixing conditions at that time of year. Consequently, no changes in the seasonal minima are observed at all measurement sites, which is taken as an indicator of enhanced thickness of the mixing layer as good mixing conditions. Taking *ZSF* as an example, Birmili et al. (2009) observed low concentrations of particle number in winter and found it representative for the free tropospheric air by analyzing the annual and diurnal cycles. From spring on, the warmer it gets the higher the PBL goes. The intense vertical atmospheric exchange during summer months results in a daily air mass transport from the boundary layer to reach *ZSF* due to thermal convection (Reiter et al., 1986; Birmili et al., 2009). Thus there are optimal transportation and mixing conditions."

Page 11 line 6: explain in more detail 'thickness of the mixing layer'.

Please see the last answer above.

Page 11 line 14: what does 'least standard deviations' mean?

This refers to the variability in the remainder components.

Thus, we rephrased it as, "the smallest standard deviations in the remainder components."

Page 11 line 14: we already knew that IZO is least influenced.

*IZO* was taken as reference station, according to our assumption at the beginning. Therefore, as the results showed the remainder component at *IZO* with the least standard deviation, it supported our assumption that it was the least influenced station among all.

Page 11 line 15: what are intermediate results?

We rephrased as, "The three alpine measuring stations (*ZSF*, *SNB* and *JFJ*) exhibit intermediate variability."

Page 11 line 19: could, but why not done?

We deleted this sentence as it was beyond the focus of this paper.

Page 11 line 25, figure 6: why is red included in the color scale as those values do not occur? Also the caption of figure 6 contains a lot of information that should be included in the main text as well (pearson corr. matrix etc).

We changed the color scale in Figure 6 from white to blue only.

And we added the explanation of the figure in the main text, "The trend and seasonal components of all VAL and selected datasets were firstly compiled, and then Pearson's correlation coefficients were calculated assuming normal distribution of data examined by Anderson Darling test (P < 0.05). The correlation matrices are shown for each type of data sets individually. Data used for correlation were chosen only when available at all stations (2012–2015)".

Page 12 line 1: what does this mean/imply?

We rephrased the sentence as, "The number of insignificant correlations between the station pairings is the greatest for ADMV."

And we also added at the end of the paragraph, "This means that by ADMV, the combination of trend and seasonal components correlate best and the remaining unselected data have the lowest correlation among the methods. If these two criteria are used to separate the representative part of the data from the unrepresentative part, the ADMV method produces the best results".

Page 12 conclusions: should be especially checked for level of English.

An English proofreading has been done throughout the paper.

Page 12 line 7: not all 6 cover this period.

We deleted "from 2010 to 2016."

Page 12 line 7: rewrite, ABF does not select..

We replaced by, "The ADMV selection resulted in…"

Page 12 line 8: growing elevation?

We deleted "growing."

Page 12 line 10: but what does it mean/imply that is the most restrictive? When would you recommend the ABF method?

ADMV is the most restrictive in terms of selection percentage, which selects the least data representative for the lower free troposphere. However, additional indicators should be defined for the selection quality, such as STL and correlation analysis mentioned in the content.

Regarding the results of correlation analysis, we would recommend ADMV for selection of representative well-mixed lower free tropospheric air for the elevated mountain stations mentioned in this study.

Page 12 line 14: what do reduced and delayed mean here?

We rephrased "reduced and delayed influences of $CO_2$ sources and sinks" as, "with smaller seasonal amplitudes and delayed occurrences of seasonal maxima."

Page 12 line 18-19: what do you mean?

We rewrote this paragraph as the following.

"The presented method ADMV is useful for data selection of atmospheric $CO_2$ data representative of the lower free troposphere. It requires only data from a single measurement site. It is easily adjustable to the local conditions and it runs automatically. The method can also be applied on historical datasets. The results provide evidence that the proposed ADMV method confers the possibility of selecting data that are representative of $CO_2$ concentrations of a larger area of the lower free troposphere. This is an elementary prerequisite for application of the method to a large number of different stations and an essential step toward generalization. It directly supports the objective of GAW to extrapolate from a set of point measurements from single

stations to a larger representative area or region in the lower free troposphere (WMO, 2017b). In future, there is a need to test whether such results could be used for additional tasks, such as ground calibration of satellite measurements".

Page 12 line 21: how applicable is the method to other stations?
    This would be one of our next research questions and would be tested in the near future.

Figure 1: add larger map to know which region of the world this is.
    This was corrected.

**Reference**

Birmili, W., Ries, L., Sohmer, R., Anastou, A., Sonntag, A., König, K., and Levin, I.: Feine und ultrafeine Aerosolpartikeln an der GAW-Station Schneefernerhaus/Zugspitze, Gefahrst. Reinhalt. L, 69, 1/2, 31–35, 2009.

Henne, S., Brunner, D., Folini, D., Solberg, S., Klausen, J., and Buchmann, B.: Assessment of parameters describing representativeness of air quality in-situ measurement sites, Atmos. Chem. Phys., 10, 3561–3581, doi:10.5194/acp-10-3561-2010, 2010.

Herrmann, E., Weingartner, E., Henne, S., Vuilleumier, L., Bukowiecki, N., Steinbacher, Coen, F., Collaud Conen, M., Hammer, E., Jurányi, Z., Baltensperger, U., and Gysel, M.: Analysis of long-term aerosol size distribution data from Jungfraujoch with emphasis on free tropospheric conditions, cloud influence, and air mass transport, J. Geophys. Res., 120, doi 10.1002/2015JD023660, 2015.

Reiter, R., Sladkovic, R., Kanter, H.-J., 1986. Concentration of trace gases in the lower troposphere, simultaneously recorded at neighboring mountain stations. Meteorl. Atmos. Phys. 35 (4), 187–200. 10.1007/BF01041811.

Satar, E., Berhanu, T. A., Brunner, D., Henne, S., and Leuenberger, M.: Continuous CO2/CH4/CO measurements (2012–2014) at Beromünster tall tower station in Switzerland, Biogeosciences, 13, 2623–2635, doi:10.5194/bg-13-2623-2016, 2016.

Sirignano, C., Neubert, R. E., Rödenbeck, C., and J. Meijer, H. A.: Atmospheric oxygen and carbon dioxide observations from two European coastal stations 2000-2005: continental influence, trend changes and APO climatology, Atmospheric Chemistry and Physics, 10, 1599–1615, 2010.

Uglietti, C., Leuenberger, M., and Brunner, D.: European source and sink areas of CO2 retrieved from Lagrangian transport model interpretation of combined O2 and CO2 measurements at the high alpine research station Jungfraujoch, Atmospheric Chemistry and Physics, 11, 8017–8036, doi:10.5194/acp-11-8017-2011, 2011.

WMO: WMO Greenhouse Gas Bulletin, No. 13, ISSN 2078-0796, 30 October, 2017a.

WMO: WMO Global Atmosphere Watch (GAW) Implementation Plan: 2016-2023, 2017b.

---

## Author Comment (AC2) · 5 Dec 2017

**Ye Yuan et al., Adaptive Baseline Finder, a statistical data selection strategy to identify atmospheric CO$_2$ baseline levels and its application to European elevated mountain stations**

Answers to J. Kim (Referee)

The referee's comments are in black, answers are in blue.

**Short notice:**

According to the suggestion from J. Kim (Referee #2), we changed the name of our method "Adaptive Baseline Finder (ABF)" into "Adaptive Diurnal Minimum Variation (ADMV)". All the names and abbreviations of this method have been adjusted throughout the answer.

This work presents a new statistical algorithm, named ABF, for identifying "baseline" levels from CO2 measurements. The title of the work refers to elevated mountain sites as its application focus, but the work also includes some analysis of non-mountain sites as well. While there are some issues that I would like to see the authors address, overall I do feel the authors have done a good job of presenting a unique algorithm and comparing it to other frequently used methods in the measurement community, and as such I suggest that the manuscript be published with some revisions.

Before I proceed with my comments on the paper, I would like to comment on the term "baseline" itself. My concern is that the definition of "baseline" is very subjective open to interpretation. For example the authors mention that ABF in this study was used specifically to identify periods of free troposphere concentrations in the high elevation sites, and that is certainly one valid definition of "baseline". With this definition, however, sites that may have statistically stable concentrations at certain times of the day but do not necessarily measure the free troposphere will by definition have no "baseline". If the definition of "baseline" was "typical concentrations you would probably measure at a certain location at a certain time" with the goal of creating a global spatial map of average concentrations, I suppose you would end up with something close to the trend and seasonal components in the STL analysis, which you may (or may not) be able to find through statistical methods such as ABF. On the other end of the spectrum, for a regional modeler, the useful definition of "baseline" would be whatever concentrations enter the modeling domain and not necessarily any clean/stable condition, and if the air was polluted coming into the grid box then the model needs to know about it. I've seen attempts to distinguish between "baseline" and "background" to try to navigate through the subtle (and sometimes not-so-subtle) differences in definitions, but in my view all attempts at defining "baseline" is inherently subjective and the best practice is to be specific about what the particular definition for the study is, and that definition should encompass the specific intended use of this definition. All this to say, I feel the name Adaptive BASELINE Finder, while sounding nice, can be misleading. I would suggest that the authors consider another name, but will leave the decision to the authors.

Thank you very much for your concerns and elaboration. We fully agree with the elaborate remark of the reviewer. We would like to recall that our study largely focuses on elevated stations, but we define the term "baseline" in the beginning of the introduction (see Page 2 line 8-10) in a broader sense referring to "well-mixed air masses with minimized short-term external influences." We also agree that the name "Baseline Finder" can be misleading, thus after discussion with all co-authors, we decided to change the name into Adaptive Diurnal Minimum Variation (abbreviated: ADMV). Besides, we also added more arguments mainly in the Results and

discussion, as well as the Conclusion section.

**[General comments, questions]**

- P5, ln 15: Why the window of 6 hours? I suppose this assumes that baseline conditions occur for longer than 6 hours? Have you tried shorter windows and found you come to the same conclusion? I almost wonder whether it would be more beneficial as a general algorithm to have as short a window as possible, such that the window never exceeds the actual window of a baseline occurrence?

The window of 6 hours and the standard deviation threshold were choices based on empirical visual inspection of the available datasets and on literature review. For this study, we specify that the 6 hours in the *start time window* have to meet two constraints: the standard deviation of measured values less than 0.3 ppm and the missing data in the 6 hours less than 50%. If the requirements are fulfilled, then the data selection will start with the *start time window* for that day. Otherwise, all values in that day with the *start time window* are labelled as "unselected."

The length of 6 hours was considered as a reasonable time length to determine whether the measured air masses are well-mixed and thus most representative, largely following the approach of Pales and Keeling (1965) (as mentioned in the introduction, page 3 line 15). More detailed information can also be found in Levin et al. (1995) and Brailsford et al. (2012). Shorter time windows will lead to less robust statistics and thus more variable standard deviations. Thus the selection procedure might rely on less representative data and risk of accepting the wrong *start time window* increases.

However it is worth noting that 6 hours were only chosen for this study. It can be variably adjusted by users according to their measurement sites.

- P10, ln 15: The increase in the mean annual growth rates is within the noise, I'm not sure that much can be made of this.

We agree that, the tendency is not statistically significant based on the confidence intervals.

We rephrased as, "Moreover, the following fact is observed for all sites except for *SSL*."

- Figure 2: I have a hard time understanding this figure. First off, the figure seems to represent data from the full data set (spanning years), and yet the method describes that the baseline "window" is adaptive, potentially changing each day and by season. What criteria was used to derive a representative window for the whole period?

We apologize for the misleading wording. There are different terms regarding time window in our ADMV method: *start time window* and *selected time window*.

The *start time window* is different based on different running frequency of ADMV. It is the result from the first part of ADMV data selection – *starting selection* (see Section 2.2.1). For this study we applied overall frequency, indicating the *start time window* for the full data set (spanning years) is the same. Figure 2 shows the *start time windows* at each measurement site. Theoretically we can also apply ADMV by yearly, seasonally or monthly depending on the requirements.

And for calculation, the *start time window* is derived from the diurnal cycles which are the mean over the detrended data. The criterion for selecting such window is the least variable time period (6 hours) during the night time (6 p. m. to 5 a. m. LT), due to the focus on mountain

stations for this study. More details can be found in Section 2.2.1 (Page 5 line 6 onwards).

On the other hand, the *selected time window* represents the selected data sets from ADMV data selection. It is the result from the second part of ADMV data selection – *adaptive selection* (see Section 2.2.2). After both *forward* and *backward adaptive selection*, the *selected time window* result is different for each individual day.

- P10, ln 27: Regarding "active vegetation", wouldn't signals from respiration also explain these results, and wouldn't that also be one form of active vegetation? I think this possibility can't be ignored since the authors suggest that the lower VAL values in summer are likely due to vegetation. Are the anthropogenic emission activities in this region such that you would expect emissions only in winter, or are they small enough to be masked by the summer drawdown? I do think that the authors' interpretations on the findings are likely to be correct, however I do think that a much deeper analysis of the data (perhaps beyond the scope of this paper) may be needed to conclusively determine the source of these discrepancies.

Based on our results, it is very likely that the lower free troposphere will respond in a delayed manner to $CO_2$ concentration changes by effective sources and sinks on the ground. The lower free troposphere acts like an atmospheric "memory" with delayed reaction.

Regarding anthropogenic emissions in summer, we agree that they are certainly small enough to be masked in the drawdown. One example can be found in Oney et al. (2017) for a comparison of biospheric and anthropogenic contribution from $CO_2$ observations at a tall tower station on the Swiss Plateau, which is the most populated and most industrialized region in Switzerland. Both Fig. 2 and Fig 3 in Oney et al. (2017) show the difference in anthropogenic and biospheric signals, especially in summer time. The magnitudes of anthropogenic contributions are much smaller than the biospheric ones.

- One discussion I think is missing is regarding the "adaptiveness" of the algorithm, in other words do the results show baseline windows changing with season. The authors state this as a strength of the ABF (P4 ln 29), so I had expected this to be one of the early points of discussion.

The adaptiveness of the algorithm is indeed the ability to select values in different time windows for every individual day. It is the ability to adapt the selection on a daily basis in order to receive a maximum amount of representative data (For more details please see Section 2.2.1 and 2.2.2). One of the results is shown in Section 3.1 for the different *start time* windows among stations.

Moreover, the ADMV data selection can also be processed for each season individually (with individual settings manually). A comparison of the resulting *start time windows* between overall and seasonal running frequencies can be found in Supplement S1.1.

**[Minor comments]**

- Page 4, ln 10: "At last", change to "Finally"?

This was corrected.

- P4, ln 27: "No upwind air masses with depleted CO2 levels by photosynthesis of vegetation like in summer are recorded." -> "Unlike summer, no upwind air masses with depleted CO2 levels by

photosynthesis of vegetation are recorded."

This was corrected.

- P5, ln 12: "but preserves of the diurnal pattern." -> "while preserving the diurnal pattern."

This was corrected.

- P6, ln 10: "Step 3" is not actually a step, but a general description of Step 5 and 6. Perhaps it makes more logical sense to include it in "Step 2", presenting it as an "If/Else" step.

We combined Step 3 into Step 2 with the "If/Else" step, and changed the following step numbers accordingly.

- P9, ln 5, Table 2: Can the authors clarify whether the percentages are based on just the time windows considered in the algorithm or the complete dataset?

The percentages refer to the ratio of the selected data values in the data values of the complete data sets. The selection percentage regarding the selected time windows and the selected days can be found in Supplement S3.1 in detail.

Thus for clarification, we rephrased "data in all data for all stations" as, "data values in all values of the complete data sets".

**Reference**

Brailsford, G. W.; Stephens, B. B.; Gomez, A. J.; Riedel, K.; Mikaloff Fletcher, S. E.; Nichol, S. E.; Manning, M. R. (2012): Long-term continuous atmospheric $CO_2$ measurements at Baring Head, New Zealand. In Atmospheric Measurement Techniques 5 (12), pp. 3109–3117. DOI: 10.5194/amt-5-3109-2012.

Levin, I.; Graul, R.; Trivett, N. B. A. (1995): Long-term observations of atmospheric $CO_2$ and carbon isotopes at continental sites in Germany. In Tellus 47B, pp. 23–34.

Oney, B., Gruber, N., Henne, S., Leuenberger, M., & Brunner, D. (2017). A CO-based method to determine the regional biospheric signal in atmospheric $CO_2$. Tellus B: Chemical and Physical Meteorology, 69(1), 1353388.

Pales, J. C. and Keeling, C. D.: The Concentration of Atmospheric Carbon Dioxide in Hawaii, J. Geophys. Res., 70, 6053–6076, doi:10.1029/JZ070i024p06053, 1965.

---

## Referee Report (RR1)

"Adaptive selection of Diurnal Minimum Variation: a statistical strategy to obtain representative atmospheric $CO_2$ data and its application to European elevated mountain stations"

The manuscript has improved in this revision, the authors have especially clarified parts of the text. The replies to my comments are also helpful, and clarify the method and some issues. However, there are a number of questions that have been answered only in the reply, and the clarifications were not included in the main text. I highlight these questions/answers below and think this should still be included in the manuscript. Besides these suggestions, I also give some comments below on the revised version.

The authors write that the manuscript was proofread for English, however, the level of English in this new version still does not seem adequate and would need to be improved.

All in all, as I wrote in the review of the first version of the manuscript, I think the work done by the authors is interesting enough to be published, after taking into account the following comments.

=======

(Some comments from the first review are repeated here in gray. Author replies are included in blue, and new comments are given in black.)

Some comments on the revised manuscript or replies to the first review:

I agree with the second reviewer that replacing the terminology "baseline" in the name of the presented method is a good idea. However, I think that the new name "Adaptive Diurnal Minimum Variation (ADMV)" is not clear and possibly not correct English, and specifically I do not understand what the authors mean by "variation". The previous choice for "finder" gave an idea of the goal of the new technique, this is now missing. In the title it is more clear, as it includes 'selection'. Please reconsider the name again, possibly by including 'selection technique' or a similar term in the full name.

Page 8 line 10: Why hourly? How did you define hourly values? As the average of the whole hour? Or just last part? Is the hour defined at the beginning of the averaging interval or at the end? This is important information and should be included in methods.
Hourly values are used because of the availability of hourly averages as the highest time resolution in the World Data Center for Greenhouse Gases (WDCGG). Therefore in order to keep the format of input data constant for ADMV method, we calculated the average of the whole hour for all data sets. The time stamp for the hourly average was defined as the beginning of the averaging interval.

Moreover, originally ADMV was developed based on 30-min time resolution at the station ZSF. Therefore ADMV method can also handle data with higher time resolution than one hour.
We added in Section 2.1, "In addition, the time stamp was defined at the beginning of the averaging interval."
The argument to use hourly data because that is what is available on WDCGG does not make sense, as for each measurement site there is a co-author included in the manuscript, suggesting that they contributed to the research. Since continuous data is available, it would have been more solid if the authors had analyzed the influence of using hourly averages, versus original data. The reply that the method works on half hourly averages in that sense does not add much information, it would be more informative to compare the outcomes, and conclude what would be the optimal time resolution of the observations. Can the authors add some information on that?

Page 11 line 10: is this referring to the global growth rate? Why would you compare to an average over the last 10 years, and not to the mean growth rate during the same time period as your data sets? You could use e.g. the annual global growth rates from
https://www.esrl.noaa.gov/gmd/ccgg/trends/global.html.

Page 13 line 16 'increasing percentage of data': explain that this is the amount of data left after selection and that it is supposed to represent background conditions. In that way it is more clear to read the conclusions without the rest of the text.

Page 13 line 19 and conclusions: 'ADMV is the most restrictive'. This should be put in perspective? Is it good or bad to have a restrictive filter? Would ADMV result in better representative conditions than the other methods?

Page 10 line 27: if VAL is all validated data it can never over- or underestimate $CO_2$ levels, as they are the actual observations!
We apologize for the misleading wording.
VAL data are validated correct measurements, adjusted to the international standard reference scales and following the Global Atmosphere Watch quality objectives. Nevertheless, due to the different time scales of transportation effects VAL data may contain values from a time period where the well mixing assumption is violated (short time events). Since we referred VAL as validated unselected measurements, the $CO_2$ levels mentioned here refer to the background level of $CO_2$ which are supposed to take place at the measurement sites.
If all validated data are used, this would result in an overestimation of the atmospheric $CO_2$, due to the dominance of anthropogenic activities and no active vegetation in winter. Thus, it indicates that the VAL data are not representative.
Part of the answer here shows that my question was misunderstood: 'If all validated data are used, this would result in an overestimation of the atmospheric $CO_2$'. This is not true, as validated data are the measurements of atmospheric $CO_2$ concentrations themselves. Yes, this includes local effects etc. and are not background levels, but it is the actual $CO_2$ concentration at that

location. Change on page 11 line 21-22 to e.g.: 'indicating that the $CO_2$ concentrations estimated by VAL are above the background levels'.

=======

In a number of places, I had raised some questions with the intention to also clarify these issues in the manuscript itself. In some cases the authors have just answered these questions in the reply, but have not updated the text accordingly. I would recommend to revise the text especially to include parts of the following replies in the text:

Page 8 line 15: Does it make sense to have different windows at the different levels?
The different start time windows at the different levels result automatically from the ADMV method. It always searches for the optimal start time window based on specific data sets. In our opinion, these are very interesting and valuable results, which reflect to some extent the different characteristics of different measurement sites and also different levels. In this respect, the different time windows at the different sampling levels are results of differences in the dynamics of atmospheric transport.

Page 8 line 19: The results ARE not fully comparable. Does it even make sense to analyze such a short record which does not even give a complete annual cycle?
We agree that the data were not fully comparable because the time period was too short in contrast to the other stations. However, the results showed that for time periods shorter than a full year, the ADMV method was still applicable to the data from the tower measurements, which highlights the flexibility of the approach.

Page 9 line 2: It would make sense to look at the differences by season, as the diurnal cycle is not the same throughout the year. Also, the data sets all cover different time periods, so it is difficult to compare.
We agree that there are differences in diurnal patterns among seasons. We also applied the ADMV method separated by season, i.e., data sets were processed and selected by the ADMV method only during a specific season over the whole time period. However, we found that the start time windows didn't differ significantly (see Supplement S1.1).
Regarding different time periods of the sites we also included data of 2015 for SSL. Now except for HPB, all the measurement sites cover the same time period.

Page 12 line 21: how applicable is the method to other stations?
This would be one of our next research questions and would be tested in the near future.

---

## Author Response (AR2)

**Answers to**

Review of revised manuscript by Yuan et al.

"Adaptive selection of Diurnal Minimum Variation: a statistical strategy to obtain representative atmospheric $CO_2$ data and its application to European elevated mountain stations"

We would like to thank the referee again for reading our comments carefully and giving comments. Answers are given under each new comment, and the corresponding changes are made in the main text. Moreover, slight changes regarding affiliations, acknowledgement, and level of English were made.

We keep the color code as it is in the review. Some comments from the first review are repeated here in gray. Author replies are included in blue, and new comments are given in black.

The new replies are shown in red.
* * *
The manuscript has improved in this revision, the authors have especially clarified parts of the text. The replies to my comments are also helpful, and clarify the method and some issues. However, there are a number of questions that have been answered only in the reply, and the clarifications were not included in the main text. I highlight these questions/answers below and think this should still be included in the manuscript. Besides these suggestions, I also give some comments below on the revised version.

Thank you for your comment. We have carefully checked these questions/answers listed below and included changes accordingly in the main text. Details can be seen later.

The authors write that the manuscript was proofread for English, however, the level of English in this new version still does not seem adequate and would need to be improved.

Thank you for your suggestion. We have gone through the main text again carefully and tried to improve the English. Certain changes can be seen in the main text.

All in all, as I wrote in the review of the first version of the manuscript, I think the work done by the authors is interesting enough to be published, after taking into account the following comments.

Thank you for your kind comment.
* * *
Some comments on the revised manuscript or replies to the first review:

I agree with the second reviewer that replacing the terminology "baseline" in the name of the presented method is a good idea. However, I think that the new name "Adaptive Diurnal Minimum Variation (ADMV)" is not clear and possibly not correct English, and specifically I do not understand what the authors mean by "variation". The previous choice for "finder" gave an idea of the goal of the new technique, this is now missing. In the title it is more clear, as it includes 'selection'. Please reconsider the name again, possibly by including 'selection technique' or a similar term in the full name.

Thank you very much for your suggestion. After discussion, we changed the name to "Adaptive Diurnal least Variation Selection (ADVS)". We added "selection" to the name and adjusted the order of phrasing. Also, "variation" in the name indicates that it is a statistical measure of variation in the diurnal cycle of the time series.

Page 8 line 10: Why hourly? How did you define hourly values? As the average of the whole hour? Or just last part? Is the hour defined at the beginning of the averaging interval or at the end? This is important information and should be included in methods.

Hourly values are used because of the availability of hourly averages as the highest time resolution in the World Data Center for Greenhouse Gases (WDCGG). Therefore in order to keep the format of input data constant for ADMV method, we calculated the average of the whole hour for all data sets. The time stamp for the hourly average was defined as the beginning of the averaging interval.
Moreover, originally ADMV was developed based on 30-min time resolution at the station ZSF. Therefore ADMV method can also handle data with higher time resolution than one hour.
We added in Section 2.1, "In addition, the time stamp was defined at the beginning of the averaging interval."

The argument to use hourly data because that is what is available on WDCGG does not make sense, as for each measurement site there is a co-author included in the manuscript, suggesting that they contributed to the research. Since continuous data is available, it would have been more solid if the authors had analyzed the influence of using hourly averages, versus original data. The reply that the method works on half hourly averages in that sense does not add much information, it would be more informative to compare the outcomes, and conclude what would be the optimal time resolution of the observations. Can the authors add some information on that?

Thank you very much for your comment. The purpose of using hourly averages based on the availability on WDCGG is that, it is easily applicable for most of the users with open data (most frequently as hourly data) to apply our data selection method. A central purpose is to design an applicable methodology for Global Atmosphere Watch. And the worldwide GAW Database has no finer time resolution than 60 minutes.
Moreover, we agree that analyzing the influence of hourly averages versus original data would make a difference. However, the original time resolutions of validated data sets we got from different stations are different (from seconds, 30-min, to hourly). Therefore it is difficult to perform an overall analysis for all stations. We have included such comparisons at 2 sites in the supplement (S1.3 Time resolution $tr$). We compared 30-min time resolution with hourly averages at ZSF, showing a significant difference in the resulted selection percentages. Then we compared the data sets at JFJ by using time resolution of 10-min, 20-min, 30-min, and hourly. As a result, it follows the same pattern that, the higher the time resolution is, the lower the selection percentage is. This can be explained by the statistical property of our data selection method focusing on the variation of data under a certain time period. Therefore, with more data (higher time resolution) in the same time period, more variation could be derived so that less data can be selected based on the same threshold criteria.

As a result, the use of higher time resolution requires more strict control of the measurement process and better measurement data quality, in order to derive of the similar level of selection percentages. As a balance between selection percentage and time resolution regarding applicability, we would still recommend the use of hourly averages.

Therefore, we changed the following sentence on Page 4 line 20 as, "For this study unless otherwise indicated, hourly data were used consistently for the purpose of evaluating the data selection method as practical as possible."

Page 11 line 10: is this referring to the global growth rate? Why would you compare to an average over the last 10 years, and not to the mean growth rate during the same time period as your data sets? You could use e.g. the annual global growth rates from
https://www.esrl.noaa.gov/gmd/ccgg/trends/global.html.

Thank you very much for your suggestion. We recalculated the mean annual global $CO_2$ growth rates from 2010 to 2015 based on the data from https://www.esrl.noaa.gov/gmd/ccgg/trends/global.html for comparison in the main text now. Then we changed the sentence as, "the mean annual growth rate of *IZO* agrees well with the mean annual global $CO_2$ growth rates (2.31 ppm) during the same time period (2010-2015) based on data from https://www.esrl.noaa.gov/gmd/ccgg/trends/global.html."

Page 13 line 16 'increasing percentage of data': explain that this is the amount of data left after selection and that it is supposed to represent background conditions. In that way it is more clear to read the conclusions without the rest of the text.

We rewrote the sentence as, "The ADMV method resulted in an increasing number of selection percentages representing the background conditions with growing altitude of measurement sites, which is reasonable due to the underlying atmospheric dynamics."

Page 13 line 19 and conclusions: 'ADMV is the most restrictive'. This should be put in perspective? Is it good or bad to have a restrictive filter? Would ADMV result in better representative conditions than the other methods?

When we express, that ADMV is the most restrictive in accepting data as being representative compared with the other methods in this comparison, it cannot be argued with good or bad a priori. Rather, we face a problem in two respects.

The comparatively low yield results from the physical property of a good mixing of the lower free troposphere. As such, however, it is not good or bad, but only shows the variation of a measured physical property.

Of course, in the sense of scientific practice, it is always the aim of an investigation to generate as much representative data as possible. But what would be the point if a larger amount of data could not be used to explain the conditions in the lower free troposphere? Therefore, it is better to obtain only a comparatively low yield of representative data if it is better suited to draw conclusions about the

conditions in the lower free troposphere. At the end, the results of the correlation analysis are significant, both for the data recognized as representative and for the residuals that are rejected as not representative.

Page 10 line 27: if VAL is all validated data it can never over- or underestimate $CO_2$ levels, as they are the actual observations!

We apologize for the misleading wording.

VAL data are validated correct measurements, adjusted to the international standard reference scales and following the Global Atmosphere Watch quality objectives. Nevertheless, due to the different time scales of transportation effects VAL data may contain values from a time period where the well mixing assumption is violated (short time events). Since we referred VAL as validated unselected measurements, the $CO_2$ levels mentioned here refer to the background level of $CO_2$ which are supposed to take place at the measurement sites.

If all validated data are used, this would result in an overestimation of the atmospheric $CO_2$, due to the dominance of anthropogenic activities and no active vegetation in winter. Thus, it indicates that the VAL data are not representative.

Part of the answer here shows that my question was misunderstood: 'If all validated data are used, this would result in an overestimation of the atmospheric $CO_2$'. This is not true, as validated data are the measurements of atmospheric $CO_2$ concentrations themselves. Yes, this includes local effects etc. and are not background levels, but it is the actual $CO_2$ concentration at that location. Change on page 11 line 21-22 to e.g.: 'indicating that the $CO_2$ concentrations estimated by VAL are above the background levels'.

Sorry for the misunderstanding, and thank you very much for your clarification.

This was corrected in main text as, "When taking a closer look at the monthly averages, lower $CO_2$ values are found in the selected datasets in the winter months from October to April, indicating that the $CO_2$ concentrations estimated by VAL are above the background levels because of more dominant anthropogenic activities and no active vegetation."

In a number of places, I had raised some questions with the intention to also clarify these issues in the manuscript itself. In some cases the authors have just answered these questions in the reply, but have not updated the text accordingly. I would recommend to revise the text especially to include parts of the following replies in the text:

Page 8 line 15: Does it make sense to have different windows at the different levels?

The different start time windows at the different levels result automatically from the ADMV method. It always searches for the optimal start time window based on specific data sets. In our opinion, these are very interesting and valuable results, which reflect to some extent the different characteristics of

different measurement sites and also different levels. In this respect, the different time windows at the different sampling levels are results of differences in the dynamics of atmospheric transport.

We rewrote the corresponding sentences in the main text on page 8 line 28 as, "The observed differences in the *start time windows,* as well as in the widths of the confidence intervals (gray shades), reflect the characteristics of differently situated measurement sites and different sampling levels. The first subplot column (*HPB50*, *HPB93*, and *HPB131*), representing the three sampling heights at *HPB*, shows similar detrended diurnal patterns with similar *start time windows*. The slightly different *start time window* at *HPB131* potentially indicates different dynamics of the atmospheric transport at higher elevation."

Page 8 line 19: The results ARE not fully comparable. Does it even make sense to analyze such a short record which does not even give a complete annual cycle?

We agree that the data were not fully comparable because the time period was too short in contrast to the other stations. However, the results showed that for time periods shorter than a full year, the ADMV method was still applicable to the data from the tower measurements, which highlights the flexibility of the approach.

We rewrote the corresponding sentences in the main text on page 9 line 5 as, "The results may not be fully comparable, but instead it shows that the data selection method is also applicable to data with time periods shorter than one year."

Page 9 line 2: It would make sense to look at the differences by season, as the diurnal cycle is not the same throughout the year. Also, the data sets all cover different time periods, so it is difficult to compare.

We agree that there are differences in diurnal patterns among seasons. We also applied the ADMV method separated by season, i.e., data sets were processed and selected by the ADMV method only during a specific season over the whole time period. However, we found that the start time windows didn't differ significantly (see Supplement S1.1).

Regarding different time periods of the sites we also included data of 2015 for SSL. Now except for HPB, all the measurement sites cover the same time period.

We understand this comment. And we have made such comparisons of *start time windows* by seasons in Supplement S1.1. And in order to clarify this problem as early as possible, we made such an argument in the method section on page 6 line 10 as, "Being aware that calculating the *start time window* from overall data could differ from the *start time windows* calculated by season, the overall generated *start time windows* have been compared with seasonally generated *start time windows* for high elevated mountain stations (see Supplement S1.1). Because these differences were mostly minimal to moderate and this work aims at a methodical comparison under identical conditions, the *start time windows* are always derived from overall data."

Page 12 line 21: how applicable is the method to other stations?

This would be one of our next research questions and would be tested in the near future.

[revised manuscript text omitted]

---

## Author Response (AR3)

**Author's Response:**

Dear Mr. Dominik Brunner,

Thank you very much for your comments and suggestions for this manuscript. We appreciate it very much. All the suggestions have been included. And the following answers are made (in blue) regarding the remarks in the comments.

1.  Title: The new title of the method "Adaptive selection of diurnal least variation" sounds less correct to me than the previous title. Even though I am not a native English speaker, I am quite sure that the adjective "least" can not be used in this way. A variation can be small but not less, and thus it can be smallest but not least. Actually, I don't think that anything was wrong with the previous title " Adaptive Selection of Diurnal Minimum Variation", but rather that the method was later called "Adaptive Diurnal Minimum Variation" which sounds like the "variation" would be "adaptive" rather than the "selection". I would propose to call the method "Adapative Selection of Diurnal Minimum Variation (ASDMV)" or, closer to the present name, "Adaptive Diurnal minimum Variation Selection (ADVS)".

Thank you very much for your explanation and suggestion. We decide to call the method as, "Adaptive Diurnal minimum Variation Selection (ADVS)" throughout the manuscript.

2.  Please note that it is standard practice to present validated rather than raw data, and therefore the emphasis on "validated data" in this manuscript is quite confusing. Furthermore, I would refer to such data as "calibrated and quality controlled" rather than "validated", since validation typically requires comparison against independent data.
    "Data selection", on the other hand, is a very general term that is not necessarily associated with baseline data selection. I am, therefore, not particularly happy with the usage of the terms "validated data" and "selected data" in this manuscript as they are prone to confusion, but since the terms are introduced at the beginning, it is probably ok but certainly not ideal.

Thank you very much for your explanation and concern. We decide to keep the terms "validated data" and "selected data" in the manuscript, and frequently show the focus on "baseline data selection" as you suggested in your comments.

3.  Since THO requires data a sub-hourly resolution (or a measure of subhourly variability), this method was probably applied only to a subset of data. If that's the case, this should probably be mentioned here.

We still keep the first step of examining the within-hour variability for THO in case of evaluating data with higher time resolution. For hourly data, this step is automatically skipped

for the R programming routine.
Therefore, we added a sentence here, "For the hourly data used in this study, the within-hour variability is not applicable so that the first step is skipped."

4. What do you mean by "external influences"? External of what?
The main things that remain when filtering trend and seasonal components are (i) synoptic variability, (ii) diurnal variability (not relevant because your method esssentially removes diurnal variations), and (iii) local influences. I would not call any of these components "random". You may say that the remainder component "resembles random noise" since it is not correlated between sites, but it is certainly not a random signal by origin (except for the random noise of the instrument which should be quite negligible in hourly values).

We completely agree. We used "external influences" here defined as the synoptic variability in combination with local influences. And we used "random noise" here for the structural characteristics of the residues (or remainder).
Therefore, we combine the first two sentences as, "The remainder component resembles by its structure to random noise from local influences, being basically different from site to site and statistically uncorrelated with the general signal of $CO_2$ concentrations in the lower free troposphere."

5. I don't understand the colorscales in Figure 6. They seem to be useless and should probably be removed.

The color scale in Figure 6 is removed.

6. "data selection method" is way too general!!You need to mention the purpose of the data selection in this sentence, e.g. "baseline data selection method" or a "novel statistical method for selecting representative baseline data"

We rephrase it as, "novel statistical method for selecting representative baseline data."

7. you need to briefly mention the characteristics that were consistent. Otherwise this sentence says nothing.

We rewrite this sentence and the previous one as, "The ADVS method resulted in an increasing number of percentages of selected data representing the background conditions with growing altitude of continental measurement sites, which is reasonable due to the underlying atmospheric dynamics. For comparison, three well-known statistical data selection methods were applied to the same datasets and most methods yielded similar increasing percentages

with growing altitude."

8. The conclusions section should be understandable without reading the rest of the text. The reader will likely not understand the meaning of "validated" versus "selected" data in this section.

We change "all validated and selected datasets" to "all datasets before and after data selection", and adjust "the validated datasets" to "the datasets before selection" in the next sentence.

9. You need to explain what was correlated against what, since otherwise it is impossible to understand the significance of this paragraph.

We rewrite the sentence as, "For the combination of trend and seasonal components by STL, higher correlation coefficients between stations were found with ADVS data selection than SI and THO. Inversely, ADVS resulted in lower correlation coefficients between stations in the remainder components than the other methods. Both indicate a better performance of selecting baseline data by ADVS."